# SEMCODER: Training Code Language Models with Comprehensive Semantics Reasoning

**Yangruibo Ding**
Columbia University
yrbding@cs.columbia.edu

**Jinjun Peng**
Columbia University
jinjun@cs.columbia.edu

**Marcus J. Min**
Columbia University
jm5025@columbia.edu

**Gail Kaiser**
Columbia University
kaiser@cs.columbia.edu

**Junfeng Yang**
Columbia University
junfeng@cs.columbia.edu

**Baishakhi Ray**
Columbia University
rayb@cs.columbia.edu

## Abstract

Code Large Language Models (Code LLMs) have excelled at tasks like code completion but often miss deeper semantics such as execution effects and dynamic states. This paper aims to bridge the gap between Code LLMs' reliance on static text data and the need for semantic understanding for complex tasks like debugging and program repair. We introduce a novel strategy, *monologue reasoning*, to train Code LLMs to reason comprehensive semantics, encompassing high-level functional descriptions, local execution effects of individual statements, and overall input/output behavior, thereby linking static code text with dynamic execution states. We begin by collecting PYX, a clean Python corpus of fully executable code samples with functional descriptions and test cases. We propose training Code LLMs not only to write code but also to understand code semantics by reasoning about key properties, constraints, and execution behaviors using natural language, mimicking human verbal debugging, i.e., rubber-duck debugging. This approach led to the development of SEMCODER, a Code LLM with only 6.7B parameters, which shows competitive performance with GPT-3.5-turbo on code generation and execution reasoning tasks. SEMCODER achieves 79.3% on HumanEval (GPT-3.5-turbo: 76.8%), 63.6% on CRUXEval-I (GPT-3.5-turbo: 50.3%), and 63.9% on CRUXEval-O (GPT-3.5-turbo: 59.0%). We also study the effectiveness of SEMCODER's monologue-style execution reasoning compared to concrete scratchpad reasoning, showing that our approach integrates semantics from multiple dimensions more smoothly. Finally, we demonstrate the potential of applying learned semantics to improve Code LLMs' debugging and self-refining capabilities. Our data, code, and models are available at: https://github.com/ARiSE-Lab/SemCoder.

## 1 Introduction

Recent advancements in code language models (Code LMs) [1–5] have revolutionized the field of programming [6–8]. These models, trained primarily on vast corpora of programming-related text such as source code and docstrings [9], excel at automating tasks like code generation.

Unfortunately, the reliance on static text data limits the ability of existing Code LMs to understand what the programs are actually doing, especially to reason about the deeper semantics intrinsic to code execution. The lack of semantic understanding unsurprisingly often leads to poor performance in debugging and repairing errors in generated code [10]. Code LMs struggle with reasoning about program semantics in both static and dynamic settings. In a static setting, the challenge lies in understanding the intended behavior of the code without running it, requiring deep comprehension of

38th Conference on Neural Information Processing Systems (NeurIPS 2024).

code syntax and static semantic properties (e.g., program dependency graph, etc.) [11, 12]. A dynamic setting involves observing and interpreting the code's behavior during execution, including tracking variable changes, identifying runtime errors, and detecting performance issues [13]. Even when the execution traces are exposed to the model, [13] observed that Code LMs could not effectively interact with the real executions, struggling to leverage the dynamic execution traces for debugging.

Fifty years ago, Terry Winograd envisioned the future AI programmer: "The key to future programming lies in systems which *understand what they are doing* [14]". In this paper, we explore constructing such a programming system, backed up by language models, not only to write programs but also to understand what they are doing (a.k.a., semantics). Our key insight is that Code LMs should mimic how pragmatic human developers work: starting with general specifications, breaking them down into sub-tasks with expected properties and constraints, implementing code line by line while reasoning about the effects of each line, and checking overall correctness by examining execution effects [15]. To achieve this, we introduce a novel strategy to train Code LMs to reason comprehensive program semantics.

We train SEMCODER, a novel semantic-aware Code LM. We incorporate different modalities of program semantics: (i) High-Level Functional Descriptions: We train SEMCODER to understand high-level functional descriptions bi-directionally by both generating code from natural language and summarizing code as natural language. This involves teaching models to grasp a program's purpose, akin to how a human developer outlines software high-level approximate semantics; (ii) Key Properties and Constraints: we train SEMCODER to extract the functional properties and constraints of a program, which should hold for all scenarios and corner cases. (iii) Overall Execution Behavior: we train SEMCODER to understand the local impact of individual code statements, recognizing how each line affects variables, control flow, and memory usage. By grasping these effects, models can better predict code execution semantics. We train the model to learn both abstract and concrete semantics, teaching it the general purpose of a statement and illustrating it with concrete examples.

**Curating Executable Code Dataset** We collect PYX, a synthetic dataset capturing comprehensive program semantics with executable code samples and unit tests. Inspired by existing datasets [16, 17], we use a powerful LLM to synthesize NL-to-code pairs. To ensure quality, PYX includes only executable samples. It also generates unit tests and detailed execution traces, recording program states after each statement. From PYX, we further construct a debugging dataset, PYX-R. PYX-R includes buggy code snippets generated by Code LMs, corresponding debugging rationales, and refine plans [13] leading to patches. By fine-tuning Code LMs on PYX-R, we aim to develop programming assistants that debug and patch faulty code in a human-like manner, advancing the capabilities of current Code LMs in iterative programming.

**Learning Program Semantics** To learn program semantics, we propose *monologue reasoning*: Code LMs try to understand and explain the code semantics to themselves. Code LMs will summarize the program functionalities, highlight the key properties and constraints, and reason code execution step-by-step, inspired by rubber duck debugging [15]. The code execution reasoning will be performed in two directions: (i) forward monologue: SEMCODER uses source code and inputs to verbally simulate execution, explaining each line's impact, executed lines, variable changes, and final output, and (ii) backward monologue: given the final output, SEMCODER reasons about possible previous states abstractly, capturing essential characteristics without precise enumeration. This abstract reasoning is crucial for understanding complex operations like sorting or aggregation, where the previous state cannot be uniquely determined. Overall, monologue reasoning equips Code LMs with a human-like understanding of control flow, state transitions, and complex operations, bridging the gap between static code analysis and dynamic execution reasoning.

We show that, by training on this approach, SEMCODER can generate, reason about execution, debug and refine code in a more intuitive and effective manner, pushing the boundaries of what current Code LMs can achieve in different software engineering tasks.

**Performance of SEMCODER** SEMCODER, while having only 6.7B parameters, exhibits exceptional performance in code generation and execution reasoning tasks, surpassing larger models like GPT-3.5-turbo and various open-source models. For code generation, SEMCODER variants achieve a pass@1 of 79.3% on HumanEval [1], outperforming GPT-3.5-turbo's 76.8%, and with 27.5% on LiveCodeBench-Lite [18], outperforming GPT-3.5-turbo's 23.9%. For execution reasoning, SEMCODER variants score 63.6%, 65.1%, 61.2% on CRUXEval-I, CRUXEval-O, and LiveCodeBench-CodeExecution, respectively, significantly outperforming baseline models including

## ✍ Approximate Semantics

Implement a function that takes a list of potential energies, sorts them in ascending order, removes duplicates, and returns the indices of the unique sorted energies.

Write a function `unique_sorted_indices(energies: List[float]) -> List[int]` that takes a list of potential energies as input and returns a list of indices corresponding to the unique sorted energies.

## 🔧 Symbolic Semantics

```python
from typing import List
def unique_sorted_indices(energies: List[float]):
    energy_dict = {}
    for idx, energy in enumerate(energies):
        energy_dict.setdefault(energy, idx)
    sorted_unique_energies = sorted(set(energies))
    unique_sorted_indices = [energy_dict[energy] for energy in sorted_unique_energies]
    return unique_sorted_indices
```

## 🔍 Abstract Semantics

**Key Properties and Constraints**

- The input list can contain duplicate values and is of variable length.
- The output list will have unique energy values sorted in ascending order and will contain their original indices from the input list.
- If an energy value is repeated, only its first occurrence's index is stored in the dictionary.
- The function ensures that the indices in the output list reflect the order of the unique values after sorting.

## ⚙ Operational Semantics

Given the input is `[10.5, 8.2, 10.5, 7.1, 8.2]`, let's simulate the execution
1. `energy_dict` is initialized as an empty dictionary.
2. Filling the Dictionary: The `for` loop begins, iterating through the list with `enumerate`:
   - Iter 1: For `idx = 0`, `energy = 10.5`. We update the dictionary to be {10.5: 0}
   ...
   - Iter 5: For `idx = 4`, `energy = 8.2`. Since 8.2 is in the dictionary, it is not updated.
   Final state of `energy_dict`: `{10.5: 0, 8.2: 1, 7.1: 3}`.
3. Sorting Unique Energies: Resulting in `sorted_unique_energies = [7.1, 8.2, 10.5]`.
4. Retrieving Original Indices:
   - For `energy = 7.1`, the index is `3`.
   - For `energy = 8.2`, the index is `1`.
   - For `energy = 10.5`, the index is `0`.
   - Thus, `unique_sorted_indices` becomes `[3, 1, 0]`.
5. Return Statement: The function returns the list `[3, 1, 0]`.

Figure 1: SEMCODER's training strategy with different modalities of program semantics. We specify the overall objective of a task, i.e., the approximate semantics (blue box), such as "retrieves potential energies of atoms and performs sorting" followed by the corresponding code solution (pink box). Then we annotate the abstract code semantics as those key properties and constraints (red box) that hold regardless of inputs. Beyond static semantics, we also pair code with test cases, such as "Given [10.5, 8.2, 10.5, 7.1, 8.2], return [3, 1, 0]". We further annotate the dynamic, operational semantics with forward and backward monologues (yellow box, and more in Section 4.2). SEMCODER learns from all the information to not only generate code but comprehensively reason its semantics.

GPT-3.5-turbo and showcasing its superior understanding of program executions. The innovative monologue reasoning technique, where the model verbalizes code semantics from high-level functionalities to low-level execution details, greatly enhances execution reasoning, outperforming existing trace reasoning formats like scratchpad [2] and NExT [13]. The monologue reasoning approach also allows SEMCODER to flexibly handle abstract semantics and non-deterministic program states, which existing methods struggle with. Additionally, SEMCODER excels in debugging and self-refinement, improving code generation accuracy iteratively by verbally rubber-duck debugging by itself without the need for dynamic tracing. We empirically reveal that SEMCODER's static monologue reasoning is comparably effective as attaching real traces [13] for bug fixing. Besides the effectiveness, monologue reasoning has unique advantages by design: (1) it is purely static reasoning and does not require dynamic tracing, (2) it compacts the execution reasoning by focusing on key properties related to the bug rather than checking all redundant program states and concrete variable values, and (3) it provides a human-readable explanation for better understanding.

Our main contribution is the development of SEMCODER, a semantic-aware Code LM designed to enhance understanding and reasoning about program semantics. We introduce Monologue Reasoning, a novel code reasoning approach that connects static source code with its runtime behavior through detailed verbal descriptions of code properties and runtime behaviors. To expose comprehensive program semantics at different levels, we curate PYX, a collection of executable code samples with functional descriptions and execution traces. SEMCODER demonstrates superior performance in code generation and execution reasoning tasks, surpassing larger open-source models. SEMCODER also excels in debugging and self-refinement by leveraging knowledge from its semantic-aware training. Our work highlights the potential of integrating deep semantic understanding into Code LMs to improve their effectiveness in complex programming tasks.

## 2 Program Semantics

Program semantics refers to the meaning or behavior of a computer program, describing what it does when it runs, including input processing, computations, and output [19, 20]. Understanding program semantics is crucial for ensuring programs behave correctly and meet their intended purpose.

Program semantics can be represented in various modalities. A high-level description outlines a program's intended functionality, while fine-grained semantics detail the actions and side effects of each line of code, including data manipulation and state changes. This detailed understanding helps developers write better code and aids in code reviewing, debugging, and team communication. Fine-grained semantics can be concrete or abstract. Concrete semantics (e.g., program traces) capture actual execution effects, while abstract semantics focus on key input-output relationships and overall program effects, abstracting away lower-level details [21, 22]. Following the existing literature on program semantics [19, 20], we curate the following semantics.

**Approximate Semantics** describes the overall objectives of a program, often articulated through docstrings or documentation [23, 24]. These Natural Language descriptions provide an overview of the program's goals and anticipated results, ensuring that the implementation aligns with the intended high-level functionalities (blue box in Figure 1).

**Symbolic Semantics** represents complex functionality and logic in a way that both humans and machines can interpret consistently. It refers to the layer of meaning derived from the symbols, syntax, and structure of source code (pink box in Figure 1). It describes how code represents high-level functionality and logic by focusing on those constructs within the source code that symbolize particular behaviors, concepts, or operations in the program design.

**Operational Semantics** describe how the individual steps in a source code execute [25, 19, 20]. It focuses on describing the concrete execution of a program in a step-by-step manner, detailing how each action transforms the program's state. This approach is particularly useful for reasoning about the dynamic behavior of programming languages (yellow box in Figure 1).

**Abstract Semantics** is a way to describe program behavior at a higher level of abstraction [26, 27, 21, 22]. Unlike concrete semantics, which provides detailed descriptions of the program's execution on specific inputs, abstract semantics focuses on the essential aspects of program behavior while ignoring low-level details. This approach is to reason about program properties and constraints (red box in Figure 1 that always hold.

## 3 PYX: Semantic-aware Training Dataset

Capturing program semantics requires executing source code with unit tests. Real-world datasets are challenging due to diverse configurations, lack of unit tests, and limited documentation [28]. Thus, we use a synthetic dataset to capture program semantics. Here, we detail the process of gathering high-quality data for learning multi-modal code semantics. Similar to [16, 17], we first synthesize NL to Code pairs. Then, we use the Python interpreter to filter out defective samples, ensuring comprehensive semantic coverage. See Appendix F for more details and analysis, including Figure 4 which depicts the data collection procedure.

### 3.1 Synthesizing Executable Code

Synthesizing instructional data (NL to code) with existing LLMs is common for obtaining large datasets for instruction tuning CodeLLMs [16, 17]. However, current methods do not guarantee the quality of generated code. For instance, out of 43.1k Python solutions from [16], about 11.6k (26.9%) are inexecutable despite instructions to produce "correct" and "self-contained" code (Table 7 in Appendix F shows the top 10 error types). To build SEMCODER, we train it only with executable data, as good data leads to better generation [29, 30]. We improve the OSS-INSTRUCT data generation process [16], which prompts an LLM to create a programming task and solution inspired by a seed snippet. Instead of randomly sampling lines from existing programs, we parse them into ASTs and sample subtrees to obtain parsable seeds. We execute the generated code, retaining only successfully executed samples, and use the generator model's debugging capability to retry until the code runs correctly. With the low-cost supervision from the Python interpreter, we build a higher-quality instruction tuning dataset for semantic-aware model training. Step I of Figure 4 in Appendix F summarizes this process. Table 2 in Appendix F compares our PYX with OSS-INSTRUCT in details.

### 3.2 Dataset with Operational Semantics

We select a subset of PYX to construct data to learn the execution reasoning (See Step-II of Figure 4 in Appendix F).

| Forward Monologue | Backward Monologue |
|---|---|
| Given the input energies `[10.5, 8.2, 10.5, 7.1, 8.2]`, what will the function return? | Given the output is `[3, 1, 0]`, what is the function input? |
| 1. `energy_dict` is initialized as an empty dictionary.
2. Filling the Dictionary: The `for` loop begins, iterating through the list with `enumerate`:
   - Iter 1: For `idx = 0`, `energy = 10.5`. We update the dictionary to be {10.5: 0}
   ...
   - Iter 5: For `idx = 4`, `energy = 8.2`. Since 8.2 is in the dictionary, it is not updated.
   Final state of `energy_dict`: `{10.5: 0, 8.2: 1, 7.1: 3}`.
3. Sorting Unique Energies: Resulting in `sorted_unique_energies = [7.1, 8.2, 10.5]`.
4. Retrieving Original Indices:
   - For `energy = 7.1`, the index is `3`.
   - For `energy = 8.2`, the index is `1`.
   - For `energy = 10.5`, the index is `0`.
   - Thus, `unique_sorted_indices` becomes `[3, 1, 0]`.
5. Return Statement: The function returns the list `[3, 1, 0]`. | - Key constraints according to the output:
   - The first unique energy corresponds to index `3`, the second to `1`, and the third to `0`.
- Energy Values:
   - `energies[3]` -> first unique energy (let's call it `e1`)
   - `energies[1]` -> second unique energy (let's call it `e2`)
   - `energies[0]` -> third unique energy (let's call it `e3`)
- In the sorted list of unique energies, `e1` must be the smallest, followed by `e2`, and then `e3`.
- A simple set of values array can be created. We must ensure that:
   - The energy at index `3` is less than the energy at index `1` and both should be less than the energy at index `0`.
Let's select:
- `energies[3] = 1.0` (first unique, smallest value)
- `energies[1] = 3.0` (second unique, middle value)
- `energies[0] = 5.0` (third unique, largest value)
To ensure proper indexing and repetition, we can fill the rest of the list with duplicates of such values, for instance: `energies = [5.0, 3.0, 5.0, 1.0]` |

Figure 2: Forward monologue simulates the execution step-by-step, and backward monologue deduces the previous program states by making assumptions and checking with observed constraints.

**Data Selection** We apply the following filtering criteria to select programs with clean execution flow from our executable dataset: (i) Only programs without external resource interactions (e.g., keyboard input, file system changes) are included, as our trace representation only captures variable state changes. (ii) Programs must have no randomness, ensuring predictable behavior.

**Input Generation** Our executable dataset typically has one or two example inputs per program. To model operational semantics accurately and avoid bias, we need a diverse input set to expose different execution traces. We expand the input set using type-aware mutation and LLM-based input generation, similar to [31] as detailed in Appendix F.

### 3.3 PYX-R: Training Code LLMs to Rubber-duck Debug and Self-refine

We construct a debugging dataset, PYX-R, to train Code LLMs for debugging and self-refinement, aiming to improve their iterative programming capabilities. We collect buggy solutions by sampling LLM for problems in PYX and keep those responses that fail at least one of the tests. We perform rejection sampling with LLM to collect rubber-duck debugging rationales for buggy programs and their input sets. PYX-R only includes those rationales that lead to correct patches, verified by differential testing against the ground truth. We provide an example of PyX-R data in Appendix F.

## 4 SEMCODER: Learning Comprehensive Semantics

### 4.1 Natural Language to Code

We train SEMCODER to translate high-level functional descriptions into executable code, known as the natural language to code task [16, 17]. Using PYX samples, we provide well-defined problem descriptions that specify (1) the task's overall objective, (2) implementation constraints, and (3) expected outcomes with test cases. These descriptions give a holistic view of the task, forming the basis for the model's understanding.

### 4.2 Monologue Reasoning to Comprehensively Understand Code Semantics

We train SEMCODER to understand code semantics through monologue reasoning: Given the source code and executable inputs/outputs, the model needs to reason code from high-level abstraction to low-level details, from static perspective to dynamic perspective. Note that the original natural language description of the problem will not be provided to generate monologues.

First, SEMCODER summarizes the high-level functionalities to understand the approximate semantics. Then, SEMCODER will explain the abstract semantics as key properties and constraints that always hold for all executions. Finally, SEMCODER describes the operational semantics by articulating state changes during execution for the provided execution input/output. Inspired by rubber-duck debugging, this approach explains program states transition more smoothly than structured formats like Scratchpad [32], avoiding redundant program states (e.g., numpy array with hundreds of elements) and concrete values (e.g., float numbers) while focusing on key properties that contribute to the code understanding. We detail such effectiveness in Section 6.2. We provide partial monologues for illustration in Figure 2 and full monologues in Appedix G.

### 4.2.1 Forward Monologue

We provide SEMCODER with the source code and input, and it learns to reason the operational semantics by verbally simulating the execution step by step and predicting the execution output (Figure 2 yellow box).

**Execution Coverage**  To ensure comprehensive understanding, forward monologue covers those lines with side effects, contributing to a thorough control flow understanding and enforcing a detailed code walkthrough, similar to a developer's debugging process.

**Natural Execution Orders**  To mimic natural code execution, forward monologue follows the natural order of reasoning. For loops, it explains each iteration with specific values, addressing lines executed multiple times differently. This ensures an accurate, context-aware execution path, similar to how developers mentally simulate execution behavior, helping to detect issues like infinite loops or incorrect condition handling.

**Program State Transition**  Understanding code side effects is crucial for grasping program state evolution. Forward monologue indicates changes in variable values when a line is executed, enhancing its ability to simulate real execution effects. This focus on side effects helps capture dynamic semantics, providing granular, step-by-step explanations of state changes, thus improving debugging and refinement based on observed behavior.

**Final Output**  Finally, the model predicts the program's final output after explaining the execution process to validate the correctness of intermediate logic.

### 4.2.2 Backward Monologue

While forward execution is mostly deterministic, the previous program state cannot always be determined from the current state, such as an unsorted list from its sorted version. Therefore, we design the backward monologue to be flexibly abstract (See Figure 2, blue box).

**Abstract Intermediate Constraints**  In our backward monologue reasoning, we use abstract intermediate constraints when previous program states can't be uniquely determined from the current state, such as after sorting or aggregation. We train the model to describe these constraints abstractly. This abstraction captures essential characteristics and patterns, allowing the model to reason about multiple possible previous states. This approach enhances the model's flexibility and generalization, improving its ability to handle diverse and complex program reasoning tasks.

**Concrete Input**  For a given output, the model learns to predict concrete input values that satisfy the input abstract constraints. This step bridges the gap between abstract reasoning and concrete execution. This ensures it understands patterns and can generate practical examples, enhancing its robustness for real-world tasks like debugging and testing. This capability mirrors how human developers perform backward reasoning for debugging [33].

### 4.2.3 Monologue Annotation Using LLM

To annotate the monologue required for training SEMCODER, we employ a method of rejection sampling [34, 35] through a large language model. We leverage the power of LLM to automatically annotate numerous samples for training SEMCODER, while we have an execution-based golden standard to verify the quality of annotated monologues, ensuring they are informative and valuable, thereby enhancing SEMCODER's ability to reason about program executions both forward and backward.

For forward monologue annotation, we feed code samples from our PyX dataset into an LLM, prompting it to generate a detailed explanation of state changes and transition logic, ending with a final output prediction. We then execute the code; if the actual output matches the LLM's prediction, we accept the monologue, ensuring it accurately reflects the program's execution. If the output does not match, the monologue is rejected. This method ensures the monologue is comprehensive and suitable for training SEMCODER. We follow a similar strategy for backward monologue annotation.

To enhance our monologue annotation process, we provide the LLM with few-shot examples when generating forward and backward monologues. These examples follow our defined rules, explicitly detailing execution lines, variable changes, and reasoning steps for forward monologues, and abstract constraints with specific examples for backward monologues. This guidance ensures the LLM adheres

to our structured reasoning steps. We also use system instructions to ensure the LLM follows the procedures illustrated in the few-shot examples.

### 4.3 Joint Training with Comprehensive Semantics

SEMCODER is trained with the combined data of natural-language-to-code samples, forward monologues, and backward monologues, using the standard next-token prediction objective [36]. Our training has an emphasis on learning the program semantics, where the training loss is accumulated only by cross-entropy loss on code and monologue tokens together. We also include a task-specific prefix as part of the model input so that the model is better aware of which types of program semantics it should learn to capture and predict for the current sample. See Appendix H for concrete prefixes.

## 5 Experiments

**Code Generation and Execution Reasoning**    For code generation evaluation, we consider EvalPlus [31] and the code generation task in LiveCodeBench-Lite (LCB-Lite for short)[18]. For execution reasoning, we employ CRUXEval [37] and the code execution task in LiveCodeBench (LCB-Exec for short) [18]. We prompt the baseline models to perform chain-of-thought reasoning [38] motivated by two-shot examples, and zero-shot prompt SEMCODER to perform monologue reasoning. Inferences all follow the benchmark's original settings.

**Rubber-duck Debugging and Self-refine**    We evaluate iterative programming capabilities in a setting similar to self-refinement/self-debugging [39, 40] —models generate code, test it, rubber-duck debug the erroneous solution, and refine their code based on the root cause analysis. Using EvalPlus [31], we perform five iterative refinements using greedy decoding. We evaluate models with both zero-shot prompting and fine-tuned using PyX-R settings.

**Models**    SEMCODER loads the 6.7B base version of DeepSeekCoder as the initial checkpoint and continues to optimize it with the proposed program semantic training. Similar to Magicoder [16], we train two versions of SEMCODER, the base version and the more advanced SEMCODER-$S$. The base version of SEMCODER is completely trained with PYX. The advanced SEMCODER-$S$ is trained with an extended dataset that includes PYX, Evol-instruct [16], and partial CodeContest [41]. Evol-instruct is a decontaminated version of `evol-codealpaca-v1` [42], which contains numerous instruction-following data. To increase the diversity of coding problems, we sample solutions from CodeContest [41], resulting in 4.3k problems with at least one correct, LLM-generated solution.

**Configuration and Empirically Settings**    All SEMCODER variants are trained for 2 epochs on a server with eight NVIDIA RTX A6000 GPUs, using a learning rate of 5e-5 with a cosine decay to 5e-6 during the program semantics training. For self-refinement fine-tuning, SEMCODER and baseline Code LLMs are trained for 2 epochs with a learning rate of 1e-5. We use a batch size of 512, a maximum context length of 2,048. Similar to [16], we use `GPT-3.5-turbo` to synthesize coding problems. To minimize the cost, we use `GPT-4o-mini` to generate code solution and monologue reasoning texts, which are typically longer sequences than the problem descriptions.

## 6 Evaluation

### 6.1 Overall Performance

In this section, we report the overall performance of SEMCODER for code generation and execution reasoning tasks and compare it with baseline Code LLMs.

**Baselines and Evaluation Metric**    We consider four families of open-source Code LLMs as baselines: Code Llama [4], StarCoder2 [5], DeepSeekCoder [3], and Magicoder [16]. Despite SEMCODER having only 6.7B parameters, we include 6.7B, 7B, and 13B variants, both base and instruct versions, if publicly available, totaling 13 open-source models. We also compare SEMCODER to GPT-3.5-turbo for code generation and execution reasoning to measure the performance gap with closed-source models. Results are reported with pass@1.

**SEMCODER Achieves Dominant Performance in Code Generation and Execution Reasoning** We show the main evaluation results in Table 1. SEMCODER reports dominant performance in execution reasoning, significantly better than other open-source baselines, including those with

Table 1: Overall performance of SEMCODER. For code generation, the numbers outside and inside parenthesis "()" indicate the base and plus versions of EvalPlus, respectively. All results are reported with pass@1. CXEval indicates CRUXEval, and LCB indicates LiveCodeBench.

| Model | Size | Code Generation | | | Execution Reasoning | | |
|---|---|---|---|---|---|---|---|
| | | HEval (+) | MBPP (+) | LCB-Lite | CXEval-I | CXEval-O | LCB-Exec |
| GPT-3.5-Turbo | - | 76.8 (70.7) | **82.5 (69.7)** | 23.9 | 50.3 | 59.0 | 43.6 |
| CodeLlama-Python | 13B | 42.7 (38.4) | 63.5 (52.6) | 10.6 | 40.5 | 36.0 | 23.2 |
| CodeLlama-Inst | 13B | 49.4 (41.5) | 63.5 (53.4) | 12.5 | 45.6 | 41.2 | 25.7 |
| StarCoder2 | 15B | 46.3 (37.8) | 55.1 (46.1) | 16.0 | 46.9 | 46.2 | 33.6 |
| StarCoder2-Inst | 15B | 67.7 (60.4) | 78.0 (65.1) | 15.5 | 47.1 | 50.9 | 29.6 |
| CodeLlama-Python | 7B | 37.8 (35.4) | 59.5 (46.8) | 7.1 | 40.4 | 34.0 | 23.0 |
| CodeLlama-Inst | 7B | 36.0 (31.1) | 56.1 (46.6) | 10.6 | 36.0 | 36.8 | 30.7 |
| StarCoder2 | 7B | 35.4 (29.9) | 54.4 (45.6) | 11.6 | 38.2 | 34.5 | 26.3 |
| Magicoder-CL | 7B | 60.4 (55.5) | 64.2 (52.6) | 11.4 | 34.0 | 35.5 | 28.6 |
| Magicoder-$S$-CL | 7B | 70.7 (67.7) | 68.4 (56.6) | 12.1 | 42.0 | 35.8 | 30.0 |
| DeepSeekCoder | 6.7B | 47.6 (39.6) | 72.0 (58.7) | 20.3 | 39.5 | 41.2 | 36.1 |
| DeepSeekCoder-Inst | 6.7B | 73.8 (70.7) | 74.9 (65.6) | 21.1 | 41.9 | 43.2 | 34.0 |
| Magicoder-DS | 6.7B | 66.5 (60.4) | 75.4 (61.9) | 25.5 | 45.5 | 41.9 | 38.8 |
| Magicoder-$S$-DS | 6.7B | 76.8 (71.3) | 75.7 (64.4) | 23.3 | 44.6 | 43.5 | 38.4 |
| SEMCODER (Ours) | 6.7B | 73.2 (68.9) | **79.9** (65.3) | 22.4 | 62.5 | **65.1** | 59.7 |
| SEMCODER-$S$ (Ours) | 6.7B | **79.3 (74.4)** | 79.6 (**68.5**) | **27.5** | **63.6** | 63.9 | **61.2** |

$2\times$ more parameters. We also collect results for larger models (e.g., CodeLlama-34B) from the benchmark to compare with SEMCODER in Appendix Table 6.

Comparing SEMCODER with its initial checkpoint, DeepSeekCoder-6.7B, our semantic-heavy training strategy brings much stronger execution reasoning capabilities, resulting in a 23.0% absolute improvement for input prediction and 23.9% and 23.6% absolute improvement for CRUXEval-O and LCB-Exec, respectively. Notably, both variants of SEMCODER outperform GPT-3.5-turbo for execution reasoning with a significant margin.

SEMCODER also demonstrates remarkable performance in code generation: SEMCODER achieves 79.9 pass@1 in MBPP, outperforming all open-source baselines, and the advanced version SEM-CODER-$S$ achieves pass@1 of 79.3 and 74.4 for HumanEval base and plus, respectively, significantly beating other models, including GPT-3.5-turbo. These impressive results support Terry Winograd's vision in 1973 [14] that training models to thoroughly understand programs produces more reliable and accurate programming assistants.

**Execution Reasoning Requires Comprehensive Understanding of Code Semantics**   We show results of input/output prediction without reasoning in Appendix Table 5. Interestingly, when comparing the results with reasoning vs. w/o reasoning, we found that the free-form chain-of-thought can hardly help model reason about execution, even if it takes more inference-time computation to generate more tokens. In contrast, monologue reasoning significantly improves the execution reasoning capability by up to 21.7% absolute improvement in output prediction. This empirically reveals that thorough understanding of code execution requires systematic reasoning over comprehensive semantics.

## 6.2   Effectivenss of Monologue Reasoning

In this section, we perform ablation studies to demonstrate the effectiveness of monologue reasoning.

**Baselines**   We consider two baseline execution reasoning approaches: scratchpad [2] and NeXT's trace format [13]. NeXT adds numeric order to state changes and omits intermediate loop states. We also create a template to concise execution traces, replacing monologue reasoning with concrete program states. Examples are in Appendix I. Additionally, we report few-shot prompting results on the base Code LM using chain-of-thought reasoning [38] without our execution reasoning data.

**Experiments**   We first construct different formats of execution reasoning using the same PYX samples that construct monologues. Then we fine-tune `deepseek-coder-6.7b-base` on these

Table 2: Ablation study for input and output prediction with different types of execution reasoning.

| Method | CRUXEval-I | CRUXEval-O | LCB-Exec |
|---|---|---|---|
| Few-shot Prompting | 39.5 | 41.2 | 36.1 |
| Finetune | | | |
| w/ Scratchpad [2] | 48.8 | 50.6 | 39.9 |
| w/ NeXT [13] | 49.4 | 50.9 | 32.2 |
| w/ Concise Trace | 52.1 | 55.6 | 35.9 |
| w/ Monologue Reasoning (Ours) | **61.8** | **63.5** | **58.5** |

different execution reasoning data for 3 epochs and compare their results on input and output prediction using CRUXEval.

**Monologue Reasoning is More Effective Than Learning Concrete Program States**   Results in Table 2 show that, while all baselines improve execution reasoning, our monologue reasoning outperforms them in input and output prediction with clear margins. The main reason is that monologues describe state transitions smoothly in natural language while keeping track of only key properties and values, which is easier for code LLMs to learn and understand and consequently enhance execution reasoning. In contrast, baselines provide only concrete states with redundant information and values while not explaining the causal relations of these transitions, so code LLMs struggle to capture the correlation among them.

When we manually check the monologues, which are structured to ensure correct outcomes (Section 4.2.3, we observe that the intermediate logic could be occasionally flawed – the model sometimes makes wrong assumptions about code properties but still reaches the correct result. In contrast, all baselines are guaranteed to have correct intermediate steps, as they are realistic execution traces (See Appendix A for limitation and future work). Empirically, however, the model learns more effectively from the monologues. This highlights the potential benefits of emphasizing key property correctness and model-friendly data format when jointly training code LLMs with distinct semantics.

## 6.3   Debugging and Self-Refinement

We format the debugging process as verbally and statically explaining why the bug happens [15] to evaluate the code LMs' reasoning capability rather than the tool-using capability that performs dynamic execution with tracers or debuggers. Then the model should fix the bug according to its own reasoning, i.e., self-refine. We provide an example in Appendix F (Example-2) to illustrate how this task is performed.

**Experiments**   We consider four state-of-the-art instruction-tuned code LMs as baselines: Llama-3.1-Instruct-8B [43], DeepSeekCoder-Instruct-6.7B, Magicoder-DS-6.7B, and Magicoder-S-DS-6.7B. We evaluate their static debug and self-refine capabilities on EvalPlus with five iterations. We first evaluate with zero-shot prompting and then also fine-tune with PYX-R to illustrate its value.

Table 3: Performance of iterative debug and self-refine

(a) Zero-shot Prompting

| Model | Self-Refine | |
|---|---|---|
| | HEval (+) | MBPP (+) |
| Magicoder-DS | 65.2 (60.4) | 78.3 (65.9) |
| Magicoder-$S$-DS | 77.4 (70.1) | 79.9 (68.8) |
| DeepSeekCoder-Inst | 77.4 (73.2) | 80.4 (69.6) |
| Llama-3.1-Inst | 76.8 (68.9) | 77.8 (65.6) |
| SEMCODER | 75.6 (71.3) | 83.1 (67.2) |
| SEMCODER-$S$ | **84.8 (79.3)** | **86.8 (74.3)** |

(b) Fine-tuned w/ PYX-R

| Model | Self-Refine | |
|---|---|---|
| | HEval (+) | MBPP (+) |
| Magicoder-DS | 78.8 (64.3) | 83.1 (66.7) |
| Magicoder-$S$-DS | 83.5 (76.2) | 84.4 (71.4) |
| DeepSeekCoder-Inst | 83.5 (75.6) | 84.9 (69.6) |
| Llama-3.1-Inst | 76.8 (68.9) | 76.7 (61.4) |
| SEMCODER | 76.8 (69.5) | 81.7 (65.9) |
| SEMCODER-$S$ | **85.4 (79.3)** | **87.0 (73.5)** |

**SEMCODER Reports Promising Performance in Debugging and Self-Refinement**   In Table 3, SEMCODER-$S$ outperforms all baselines, more notably in the zero-shot setting. This result illustrates that the SEMCODER's monologue reasoning augments general-purpose instruction tuning with code semantics reasoning capabilities. Appendix D demonstrates SEMCODER's continuous code

refinement throughout iterations, showcasing the potential of learned program semantics for complex programming tasks.

**PYX-R Improves Iterative Programming Capability**  Fine-tuning Code LMs on PYX-R significantly improves iterative programming performance due to the monologue-style debugging rationale and well-aligned patches. PYX-R helps Code LMs understand and analyze bugs from source code and execution traces, aiming to inspire better iterative programming capabilities. We notice that PYX-R provides limited improvement to SEMCODER variants and Llama-3.1-Inst, and we speculate that these models are already trained with high-quality reasoning, and the occasional errors in PyX-R debugging rationale restrict these models from becoming significantly better (See Appendix A).

**Monologue Reasoning vs. Execution Traces for Debugging**  We perform additional experiments by replacing the monologue reasoning part (See "### Execution Simulation" in Appendix F Example 2) in the debugging rationale with real traces, following the format of NExT [13] and fine-tuning code LMs again. Results are in Appendix C.1. We notice that monologue reasoning is comparably effective as attaching execution traces. Besides the effectiveness, monologue reasoning has unique advantages by design: (1) it is purely static reasoning and does not require dynamic tracing, (2) it compacts the execution reasoning by focusing on key properties related to the bug rather than checking all redundant program states and concrete variable values, and (3) it provides a human-readable explanation for better understanding.

## 7   Related Work

**Code LLMs and Training Data**  Many open source Code LLMs, such as CodeGen [44], StarCoder [45, 5], Code Llama [4], and DeepSeek Coder [3], are proposed. Specialized models [32, 1, 41] have also been developed for tasks like code generation, summarization, output prediction, and competitive programming following the success of GPT-3 [46]. These models are trained only on source code and related text, lacking execution context. This limits their understanding of program semantics, leading to security issues and debugging failures. We aim to bridge this gap by training Code LMs on both static source code and dynamic execution traces. An orthogonal line of research curates synthetic instruction-following data to enhance Code LLM performance. Code Alpaca [47] has 20k instruction-response pairs, Evol-Instruct-Code [17] expands this to 80k pairs, and OSS-Instruct [16] includes 75k diverse pairs from the Stack dataset [9]. However, these datasets focus on natural-language-to-code tasks with little coverage of code execution and unverified solutions. To improve correctness, Zheng et al. [48] created a multi-turn conversation dataset with compiler error messages, and Wei et al. [49] incorporated execution by generating test cases and filtering invalid pairs. Yet, no dataset includes simulating and understanding execution traces. We aim to fill this gap (see Section 3).

**Learning and Reasoning about Program Executions**  Before LLMs, [50, 51] predict simple program outputs using RNNs, GNNs, small transformers, and neural Turing machines. Austin et al. [32] fine-tuned LLMs for execution output prediction with minimal performance gains. Early models predicted final outputs without revealing execution traces. Nye et al. [2] introduced the Scratchpad method for intermediate results, and others [52, 53] fine-tuned UniXcoder [54] for execution traces but didn't evaluate for code generation tasks. We fine-tune a Code LLM to understand program semantics, excelling in code generation, output prediction, and input prediction (see Section 4). Another approach uses execution feedback for debugging Code LLMs. Self-Debugging [39] shows that natural language explanations or unit test results help self-refinement, but execution traces reduce performance. LeTI [55] and CYCLE [40] fine-tune with execution feedback to improve performance, especially for smaller models. NExT [13] generates debugging rationales to mitigate the negative impact of execution traces. Our work shows that a model trained on code generation, output prediction, and input prediction excels in understanding execution feedback and self-refinement (see Table 3).

## 8   Conclusion

We train SEMCODER to simultaneously learn different modalities of program semantics: Approximate, Symbolic, Operational, and Abstract. We show that such semantics-oriented joint training cultivates a comprehensive understanding of program semantics — SEMCODER or SEMCODER-$S$ achieves SOTA performance, among all less-than-15B open-source models, in not only the code generation and input/output prediction but also tasks that require deep knowledge of both source code and execution execution reasoning like debugging and self-refinement.

## Acknowledgement

This work was supported in part by, DARPA/NIWC-Pacific N66001-21-C-4018, multiple Google Cyber NYC awards, an Columbia SEAS/EVPR Stimulus award, NSF CNS–1845995, CNS-2247370, CCF-2221943, CCF-2313055, CCF-1845893, and CCF-2107405. Any opinions, findings, conclusions, or recommendations expressed herein are those of the authors and do not necessarily reflect those of DARPA, or NSF.

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

# A   Limitations and Future Work

**Process Supervision for Intermediate Reasoning Steps**   We manually review the monologues in PYX and the rubber-duck debugging rationales in PYX-R, which are structured to ensure correct outcomes (see Section 4.2.3). While the final answers are accurate, we observed that the intermediate reasoning steps are occasionally flawed. Sometimes, the model makes incorrect assumptions about code properties but still reaches the right result, i.e., correct execution input/output and correct patch. Conceptually, such subtle mistakes in the intermediate reasoning steps might have negative impacts on further improving models' code reasoning capability.

We encourage future work to propose automatic and efficient process supervision approaches [56] specifically for code reasoning, which will be useful to further improve the quality of monologues in PYX.

**Curation of Monologue Annotation**   The monologue annotation data (Section 4.2.3) is crucial for SEMCODER to excel at the output prediction and input prediction tasks. However, we rely on a more powerful LLM, GPT-3.5-Turbo or GPT-4o-mini to generate these annotations and employ rejection sampling from its responses, since our base model is relatively small with 6.7B parameters.

We encourage future work to try our semantic-oriented joint training on a larger base model, so that it will be possible to generate the monologue annotations using the base model itself like Ni et al. [13] did to bootstrap high-quality reasoning for self-refinement.

**Incorporating Execution Reasoning into Code Generation**   We demonstrate that training on input and output prediction tasks are indirectly beneficial for both natural-language-to-code generation and downstream tasks like self-refinement. However, there is a more direct way to further improve the performance in code generation and self-refinement — we can ask the model to first self-verify its own solution by generating forward monologue (Section 4.2.1) for the test cases given in the natural language specification before finalizing the solution.

We encourage future work to explore the possibility of using a model's own execution reasoning ability to directly assist its code generation and self-refinement process.

# B   Broader Impacts

**Social Impact**   In this work, we train a semantic-aware Code LMs. We make all the data, code, and model checkpoints available publicly. The artifact could be used to deploy automated programming assistants that improve the developers' productivity. It is possible but unlikely that the Code LMs will generate buggy or wrong code, but we suggest to use our models as "copilot" to assist with human developers rather than completely relying on the model for full automation.

**Safeguards**   Our data is synthesized using a commercial LLM, i.e., GPT-3.5-turbo, which has been aligned by the releasing company, OpenAI, to avoid leaking personal or malicious information. We regard our data has minimal risk of being misused due to its synthetic instinct.

# C   More Evaluation Results

## C.1   Debug and Self-refine w/ Real Execution Traces

Table 4: Debug and self-refine with real traces in the format of NExT [13].

| Model | Self-Refine | |
|---|---|---|
| | HEval (+) | MBPP (+) |
| Magicoder-DS | 72.0 (66.5) | 83.3 (67.5) |
| Magicoder-*S*-DS | 81.7 (74.4) | 83.9 (72.0) |
| DeepSeekCoder-Inst | 84.8 (79.9) | 85.4 (70.4) |
| Llama-3.1-Inst | 76.2 (69.5) | 82.8 (66.7) |
| SEMCODER | 78.0 (70.7) | 83.6 (66.4) |
| SEMCODER-*S* | **86.0 (80.5)** | **87.0 (73.3)** |

## C.2  Input/Output Prediction Without Reasoning

In Table 5, we present the results of direct prediction for execution input and output without reasoning.

Table 5: Performance of direct prediction for execution input/output w/o reasoning.

| Model | Size | Execution Reasoning | | |
| --- | --- | --- | --- | --- |
| | | CRUXEval-I | CRUXEval-O | LCB-Exec |
| GPT-3.5-Turbo | - | 49.0 | 49.4 | 39.2 |
| CodeLlama-Python | 13B | 38.5 | 39.7 | 36.1 |
| CodeLlama-Inst | 13B | 47.5 | 40.8 | 33.8 |
| StarCoder2 | 15B | 47.2 | 46.9 | 34.7 |
| StarCoder2-Inst | 15B | 47.4 | 47.1 | 8.1 |
| CodeLlama-Python | 7B | 37.3 | 34.6 | 31.1 |
| CodeLlama-Inst | 7B | 34.8 | 35.6 | 30.1 |
| StarCoder2 | 7B | 34.2 | 35.6 | 34.0 |
| Magicoder-CL | 7B | 32.0 | 35.6 | 32.4 |
| Magicoder-$S$-CL | 7B | 36.2 | 34.8 | 30.5 |
| DeepSeekCoder | 6.7B | 42.2 | 43.6 | 44.5 |
| DeepSeekCoder-Inst | 6.7B | 34.9 | 40.8 | 41.1 |
| Magicoder-DS | 6.7B | 41.2 | 43.4 | 38.4 |
| Magicoder-$S$-DS | 6.7B | 42.1 | 44.4 | 39.2 |
| SEMCODER (Ours) | 6.7B | 46.9 | 47.9 | 38.0 |
| SEMCODER-$S$ (Ours) | 6.7B | 47.6 | 46.6 | 40.7 |

## C.3  Comparison with Larger Open-Sourced Models and Closed-Source Models

| Model | Size | Code Generation | | Execution Reasoning | |
| --- | --- | --- | --- | --- | --- |
| | | HEval (+) | MBPP (+) | CRUXEval-I | CRUXEval-O |
| GPT-3.5-Turbo-1106 | - | 76.8 (70.7) | 82.5 (69.7) | 49.0 / 50.3 | 49.4 / 59.0 |
| Claude-3-Opus | - | 82.9 (77.4) | **89.4 (73.3)** | 64.2 / 73.4 | 65.8 / **82.0** |
| GPT-4-0613 | - | 88.4 (79.3) | - | **69.8** / 75.5 | **68.7** / 77.1 |
| GPT-4-Turbo-2024-04-09 | - | **90.2 (86.6)** | - | 68.5 / **75.7** | 67.7 / **82.0** |
| CodeLlama | 34B | 51.8 (43.9) | 69.3 (56.3) | 47.2 / 50.1 | 42.4 / 43.6 |
| DeepSeekCoder | 33B | 51.2 (44.5) | - | 46.5 / - | 48.6 / - |
| DeepSeekCoder-Inst | 33B | 81.1 (75.0) | 80.4 (70.1) | 46.5 / - | 49.9 / - |
| SEMCODER (Ours) | 6.7B | 68.3 (62.2) | **79.9** (65.9) | **51.2** / 52.6 | **48.1** / 56.6 |
| SEMCODER-$S$ (Ours) | 6.7B | **81.1 (76.2)** | 78.8 (**66.9**) | 48.1 / **54.5** | 44.9 / 54.1 |

Table 6: Overall performance of SEMCODER vs. other Code LLMs. For code generation, the numbers outside and inside parenthesis "()" indicate the base and plus versions of EvalPlus, respectively. For execution reasoning, the left side and the right side of the slash "/" indicate the direct prediction and prediction with reasoning, respectively. All results are reported with pass@1.

## D  SEMCODER Continuously Refines Code Qualities

We studied SEMCODER's code generation accuracy at each step of refinement with varied temperatures. The results are plotted in Figure 3. We observed that SEMCODER is capable of continuously refining its own errors, and the increase does not stop when the temperature is high, which indicates the SEMCODER has strong debugging and self-refine capabilities and a high temperate better leverages such capabilities for iterative programming.

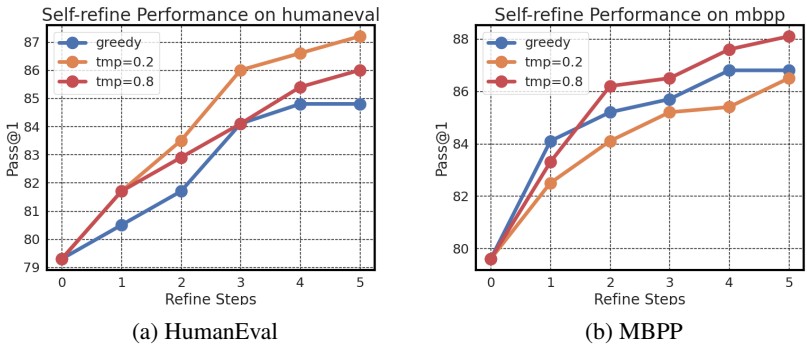

|                | (a) HumanEval | (b) MBPP |
|----------------|---------------|----------|

Figure 3: SEMCODER-S's zero-shot performance of self-refinement at each time step with different sampling strategies.

# E  Executability Analysis of OSS-INSTRUCT

Table 7: Top-10 error types of inexecutable Python code in OSS-Instruct[16]

| Error Type | #Cases out of 43.1k Python samples |
|---|---|
| ModuleNotFoundError | 3417 |
| NameError | 1954 |
| FileNotFoundError | 1052 |
| ImportError | 979 |
| EOFError | 743 |
| SyntaxError | 672 |
| IndentationError | 506 |
| AttributeError | 213 |
| TypeError | 196 |
| ValueError | 132 |

We execute all Python samples in OSS-INSTRUCT [16] to analyze their executability. To get a more accurate result, we try to mitigate the `ModuleNotFoundError` by installing the top 75 missing dependencies according to a pre-run result. Table 7 shows the breakdown of the top 10 error types.

# F  Details on PYX

The whole data collection pipeline is shown in Figure 4. Here we also document more details about PYX.

**Prompt for Data Synthesis**    We follow the prompt in OSS-INSTRUCT for data synthesis, but with two modifications: 1) For problem design, instruct the model to avoid interaction with external resources or requirement of uncommon third-party libraries to increase the probability of getting executable code. 2) For giving solutions, instruct the model to show its thought process before writing code to produce more aligned natural language along with the code in the dataset. Table 10 details our prompts with an example in PYX.

**Input Set Expansion**    To enlarge the input set, we first initialize the input corpus with all known valid inputs. Then, for type-aware mutation, we alter known inputs based on type-specific heuristics. For LLM-based generation, we prompt the model with the function and several known inputs to generate more. We verify new inputs by executing them, retaining only those that execute successfully without exceptions. We alternate between type-aware mutation and LLM-based generation until reaching a predefined threshold, combining mutation's efficiency with LLM generation's robustness. The generated inputs and their outputs serve as unit tests for the NL-described task in future steps.

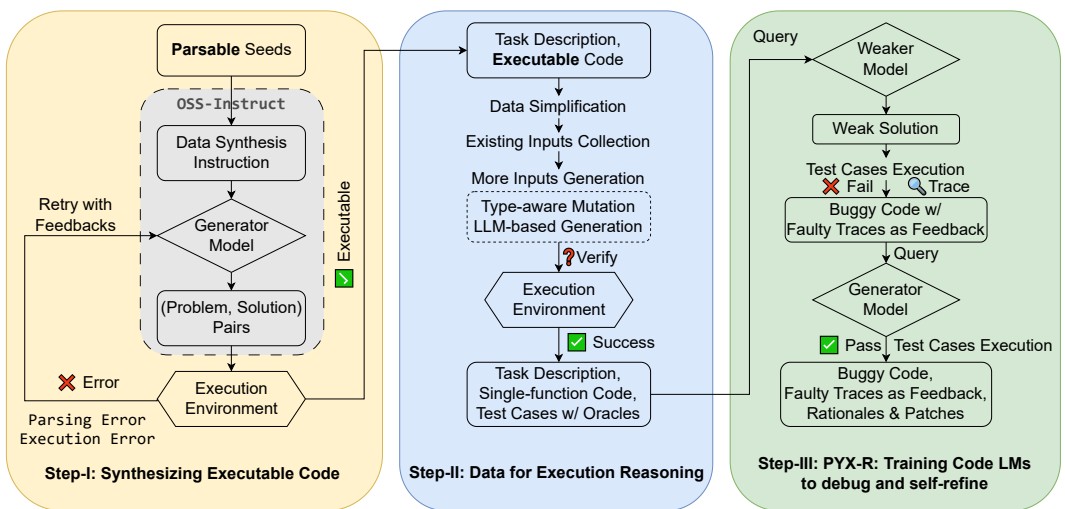

Figure 4: PYX: Execution-aware Training Data Collection Strategy

Table 8: The comparison between OSS-INSTRUCT and our PYX.

| Dataset | Problems | Seed | Solution | | Performance | |
| --- | --- | --- | --- | --- | --- | --- |
| | | Parse | Parse | Execute | HumanEval (+) | MBPP (+) |
| OSS-INSTRUCT | 75k | - | - | Partially | 67.1 (61.0) | 77.5 (64.3) |
| OSS-INSTRUCT-Python | 43k | 48% | 97% | 73% | 66.5 (59.8) | **78.6 (65.9)** |
| PYX (Ours) | 32k | 100% | 100% | 100% | **70.1 (64.0)** | **78.6** (65.1) |

**Coverage of Inputs**   Our input generation method only considers diversity in terms of variable values but does not try to fully exercise different execution paths in an executable code, like what the coverage-guided testing usually does. However, our dataset only consists of relatively short single-function programs that do not have complicated branches. We find that our generated inputs can achieve average branch coverage and average line coverage of 93%, 96% respectively, which shows that our approach is light-weight yet effective for the current setting.

**Executability v.s. Parsability**   Along with the generation of 34k executable data, we also get 13k samples that can only be parsed as valid ASTs but cannot be successfully executed. We include these parsable samples to form a 47k dataset, PYX-SOFT, and compare it against the fully executable one PYX. Results in Table 8 show that PYX-SOFT and PYX have comparable results, and both of them outperform OSS-INSTRUCT and its Python subset. Though PYX-SOFT sometimes slightly outperforms PYX, full executability is required for comprehensive semantic coverage, so we use PYX for the main evaluation.

**Data De-duplication**   We follow the data decontamination process of [16] and [57] to clean our dataset. To examine the similarity between our instruction tuning dataset and the testing benchmarks, we evaluate the "edit similarity" between them, specifically by scaling the metric `fuzz.partial_token_sort_ratio` provided by `thefuzz` library. For each sample in our dataset, its similarity to a benchmark is computed as the maximum cosine similarity to any sample in the benchmark. We apply the same analysis to OSS-INSTRUCT for comparison. Figure 5 shows that the similarity between our dataset and the two benchmarks is on par with OSS-INSTRUCT, where the majority has less than 0.4 similarity, which indicates that the performance improvement brought by our dataset is not from data leakage or benchmark data imitation.

**Categories**   To study the effect of executability filtering, we categorize all samples in our dataset following [16] shown by Figure 6. Compared to OSS-INSTRUCT, the categorical distribution shifts by an increase in algorithmic and data structure problems, data science and machine learning problems, and mathematical and computational problems, and a decrease in the remaining categories, which is expected since interactions with external resources commonly required in scenarios like database, web and UI design are not allowed in our execution environment.

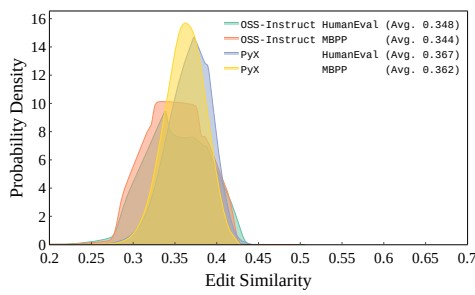

Figure 5: Edit similarities between PYX and two popular benchmarks

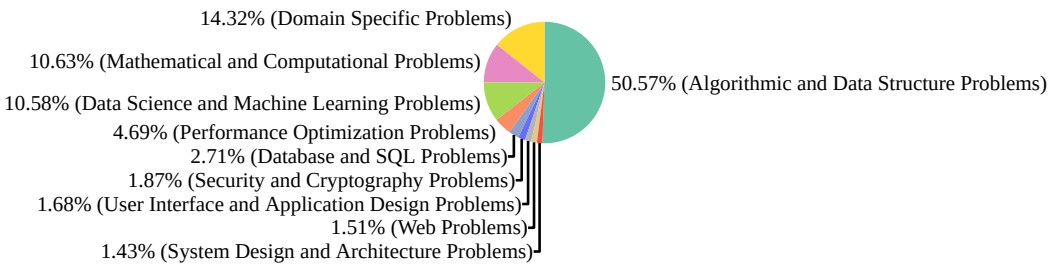

Figure 6: The category distribution of PYX

**Data Statistics** We perform decontamination on the PYX, and monologue samples (we remove the samples that share the same input and output pair with CRUXEval). PYX includes 32,489 natural language to code pairs. We generated 29,945 forward monologues and 31,022 backward monologues using rejection sampling. SEMCODER is trained with 93.4k samples, and SEMCODER-$S$ with 214.1k samples. PYX-R contains 18,473 debugging samples, each with the original description, buggy code, debugging rationale, and final patch.

---

**Example 1:** An example of PYX sample generation

**User Prompt:**

```
Please gain inspiration from the following random code snippet to create a high-quality
programming problem. Present your output in two distinct sections: [Problem Description]
and [Solution].

Code snippet for inspiration:
```

testinfra_hosts = testinfra.utils.ansible_runner.AnsibleRunner(
    os.environ["MOLECULE_INVENTORY_FILE"]
).get_hosts("instance")

def test_non_root_user(host):
    with host.sudo(user="vagrant"):
        cmd = host.run_test(command="docker container run hello-world")
    assert "Hello from Docker!" in cmd.stdout

def test_docker_group_members(host):
    content = host.file(path="/etc/group").content_string
    match = re.search(
        pattern=r"^docker:x:\d+:vagrant$",
        string=content,
        flags=re.MULTILINE,
    )
    assert match
```

```
Guidelines for each section:

1. [Problem Description]: This should be concise and **completely self-contained**,
providing all the contextual information one needs to understand and solve the problem.
Assume common programming knowledge, but ensure that any specific context, variables, or
code snippets pertinent to this problem are explicitly included. Avoid interaction with
external resources (e.g. keyboard, file system, databases, network) or requirement of
uncommon third-party libraries. The problem should be non-trivial and require some
thought to solve.

2. [Solution]: First, show your thought process and outline your planned steps clearly.
Then offer a comprehensive, **correct** and runnable solution that accurately addresses
the [Problem Description] you provided.
```

## Model Response:

[Problem Description]
Write a solution to the following coding problem:
You are tasked with creating a Python function that processes a given string to extract
specific information based on predefined patterns. The function should take a string as
input and return a dictionary containing counts of various elements found in the input
string.

The function should perform the following tasks:
1. Count the occurrences of each unique word in the input string.
2. Identify and count the occurrences of any email addresses present in the input string.
3. Determine the total count of numeric digits (0-9) in the input string.

For example, given the input string "Hello world, my email is example@email.com and my
phone number is 12345", the function should return a dictionary like this:
{
    'word_counts': {'Hello': 1, 'world': 1, 'my': 2, 'email': 1, 'is': 2, 'example@email.
com': 1, 'and': 1, 'phone': 1, 'number': 1},
    'email_count': 1,
    'digit_count': 5
}

Ensure that the function is case-insensitive when counting words and email addresses.
Words are defined as sequences of characters separated by spaces, and email addresses
follow the standard format of "example@email.com".

[Solution]
To solve this problem, we will create a Python function that processes the input string
according to the specified requirements. Here is an outline of the planned steps:
1. Initialize variables to store word counts, email count, and digit count.
2. Tokenize the input string into words.
3. Iterate through the words to count occurrences and identify email addresses.
4. Count the total number of digits in the input string.
5. Return a dictionary containing the word counts, email count, and digit count.

Here is the Python function implementing the solution:
```python

import re

def process_string(input_string):
    word_counts = {}
    email_count = 0
    digit_count = 0

    words = input_string.lower().split()

    for word in words:
        if re.match(r"[a-zA-Z]+", word):
            word_counts[word] = word_counts.get(word, 0) + 1
        if re.match(r"[a-zA-Z0-9._%+-]+@[a-zA-Z0-9.-]+\.[a-zA-Z]{2,}", word):
            email_count += 1
        digit_count += sum(1 for char in word if char.isdigit())

    return {
        'word_counts': word_counts,
        'email_count': email_count,
        'digit_count': digit_count
    }

# Test the function with an example input
input_string = "Hello world, my email is example@email.com and my phone number is 12345"
result = process_string(input_string)
print(result)
```

```

```

This function processes the input string as described in the problem and returns a
dictionary with the counts of words, email addresses, and digits. The regular expressions
used help identify words, email addresses, and digits accurately.

---

**Example 2:** An example of PYX-R rubber-duck debugging and bug fixing

---

**Prompt:**

```
### Problem
Write a solution to the following coding problem:
You are tasked with implementing a Python class that processes and represents data from
HTML soup objects. The class has three methods: `__init__`, `__str__`, and `__repr__`.
The `__init__` method takes a `soup` object, removes empty elements, and parses the input
. The `__str__` and `__repr__` methods return the object's data in JSON format.

Your task is to complete the implementation of the class by defining the `
_remove_empty_elements` and `_parse_input` methods. The `_remove_empty_elements` method
should remove any empty elements from the `soup` object, and the `_parse_input` method
should process the `soup` object to extract relevant data.

Implement the missing methods to ensure that the class functions correctly and outputs
data in JSON format when the `__str__` or `__repr__` methods are called.

### Buggy Solution
```python
from bs4 import BeautifulSoup
import json

class HtmlDataProcessor:
    def __init__(self, soup):
        self.soup = soup
        self.data = {}
        self._remove_empty_elements()
        self._parse_input()

    def _remove_empty_elements(self):
        # Remove empty tags from the soup
        for element in self.soup.find_all(True):  # True finds all tags
            if not element.get_text(strip=True):  # If the element is empty
                element.decompose()  # Remove it from the soup

    def _parse_input(self):
        # Extract relevant data from the soup
        self.data = {}
        for element in self.soup.find_all(True):  # Iterate over all tags
            if element.name in self.data:
                self.data[element.name].append(element.get_text(strip=True))
            else:
                self.data[element.name] = [element.get_text(strip=True)]

    def __str__(self):
        return json.dumps(self.data, ensure_ascii=False)

    def __repr__(self):
        return json.dumps(self.data, ensure_ascii=False)
```
### Failed Test
```python
# Test cases to verify the implementation

# Test with a simple HTML soup
html_content = "<div><p>Hello</p><p></p>World</div>"
soup = BeautifulSoup(html_content, 'html.parser')
processor = HtmlDataProcessor(soup)
assert str(processor) == '{"div": [], "p": ["Hello"], "span": ["World"]}'
```

---

**Target:**

[MONOLOGUE]
### Execution Simulation
Let's walk through the execution of the provided test case with the buggy `HtmlDataProcessor` implementation.

1. **Input HTML**: The test input is:
   ```html
   <div><p>Hello</p><p></p>World</div>
   ```
   - This gets parsed into a BeautifulSoup object `soup`.

2. **Initialize `HtmlDataProcessor`**: When `HtmlDataProcessor(soup)` is called:
   - `self.soup` is set to the parsed HTML.
   - `self.data` is initialized as an empty dictionary.
   - `_remove_empty_elements()` and `_parse_input()` are invoked sequentially.

3. **Execution of `_remove_empty_elements`**:
   - The method iterates over all elements in `soup`:
     - `<div>`: Not empty (it has children), so it remains.
     - `<p>Hello</p>`: Not empty, so it remains.
     - `<p></p>`: Empty, so it gets decomposed (removed).
     - `World`: Not empty, so it remains.
   - After this process, the `soup` now looks like:
     ```html
     <div><p>Hello</p>World</div>
     ```

4. **Execution of `_parse_input`**:
   - The method iterates over all tags again:
     - For `<div>`: It adds child elements' texts, resulting in:
       ```python
       self.data['div'] = ['HelloWorld']
       ```
     - For `<p>`: It adds its text "Hello":
       ```python
       self.data['p'] = ['Hello']
       ```
     - For ``: It adds its text "World":
       ```python
       self.data['span'] = ['World']
       ```
   - At the end of this method, `self.data` is:
     ```python
     {'div': ['HelloWorld'], 'p': ['Hello'], 'span': ['World']}
     ```

5. **String Representation**:
   - The `__str__` method is called, producing:
     ```json
     '{"div": ["HelloWorld"], "p": ["Hello"], "span": ["World"]}'
     ```

6. **Test Assertion**: The assertion `assert str(processor) == '{"div": [], "p": ["Hello"], "span": ["World"]}'` fails because the expected output for `div` is an empty list, while the generated output has `"HelloWorld"`.

### Root Cause Analysis
The primary issue lies within the `_parse_input` method. Specifically, it collects text from child elements under tags like `<div>`, which makes `self.data['div']` incorrectly aggregated with text from its children, rather than displaying children separately.

### Bug Fixing
To fix this, we should prevent the parent tag from aggregating text that includes child elements. Instead, we should only include text for elements that do not have child elements in `_parse_input`. Here's the corrected implementation:
[/MONOLOGUE]
[Refined]
```python

```
from bs4 import BeautifulSoup
import json

class HtmlDataProcessor:
    def __init__(self, soup):
        self.soup = soup
        self.data = {}
        self._remove_empty_elements()
        self._parse_input()

    def _remove_empty_elements(self):
        # Remove empty tags from the soup
        for element in self.soup.find_all(True):  # True finds all tags
            if not element.get_text(strip=True):  # If the element is empty
                element.decompose()  # Remove it from the soup

    def _parse_input(self):
        # Extract relevant data from the soup
        self.data = {}
        for element in self.soup.find_all(True):  # Iterate over all tags
            if not element.find_all(True):  # Only process elements without child
elements
                element_text = element.get_text(strip=True)
                if element.name in self.data:
                    self.data[element.name].append(element_text)
                else:
                    self.data[element.name] = [element_text]
            else:
                # If this element has children, you can initialize it with an empty list
                if element.name not in self.data:
                    self.data[element.name] = []

    def __str__(self):
        return json.dumps(self.data, ensure_ascii=False)

    def __repr__(self):
        return json.dumps(self.data, ensure_ascii=False)
```
[/Refined]

# G  Detailed Example of Monologues

We provide examples for the detailed forward and backward monologues.

**Example** Monologue Reasoning

**Code and Input:**
```
from typing import List
def unique_sorted_indices(energies: List[float]) -> List[int]:
    energy_dict = {}
    for idx, energy in enumerate(energies):
        energy_dict.setdefault(energy, idx)
    sorted_unique_energies = sorted(set(energies))
    unique_sorted_indices = [energy_dict[energy] for energy in sorted_unique_energies]
    return unique_sorted_indices
assert unique_sorted_indices([10.5, 8.2, 10.5, 7.1, 8.2]) == ??
```

**Forward Monologue:**

### Understanding the Function
The function `unique_sorted_indices` takes a list of floating-point numbers named `energies` as input. Its purpose is to find the indices of the unique elements in the input list when sorted in ascending order.
- A dictionary `energy_dict` is created to map each unique energy value to its first occurrence index.
- The `enumerate` function is used to iterate through the list and populate the dictionary with energy values as keys and their respective indices as values.
- Then, the function creates a sorted list of unique energy values from the input list using `sorted(set(energies))`.
- Finally, it creates a list of indices that correspond to these sorted unique energies using a list comprehension that retrieves the original indices from the dictionary.

### Key Properties and Constraints
- The input list can contain duplicate values and is of variable length.
- The output list will have unique energy values sorted in ascending order and will contain their original indices from the input list.
- If an energy value is repeated, only its first occurrence's index is stored in the dictionary.
- The function ensures that the indices in the output list reflect the order of the unique values after sorting.

### Execution Simulation:
Let's simulate the execution of `unique_sorted_indices([10.5, 8.2, 10.5, 7.1, 8.2])` step by step.

1. **Initialization**:
   - `energy_dict` is initialized as an empty dictionary.

2. **Filling the Dictionary**:
   - The `for` loop begins, iterating through the list with `enumerate`:
     - **Iteration 1**: For `idx = 0`, `energy = 10.5`. The dictionary is updated to `{10.5: 0}`.
     - **Iteration 2**: For `idx = 1`, `energy = 8.2`. The dictionary is updated to `{10.5: 0, 8.2: 1}`.
     - **Iteration 3**: For `idx = 2`, `energy = 10.5`. Since 10.5 already exists in the dictionary, it is not updated.
     - **Iteration 4**: For `idx = 3`, `energy = 7.1`. The dictionary is updated to `{10.5: 0, 8.2: 1, 7.1: 3}`.
     - **Iteration 5**: For `idx = 4`, `energy = 8.2`. Since 8.2 already exists in the dictionary, it is not updated.

   Final state of `energy_dict` after the loop: `{10.5: 0, 8.2: 1, 7.1: 3}`.

3. **Sorting Unique Energies**:
   - The unique energies are derived and sorted, resulting in `sorted_unique_energies = [7.1, 8.2, 10.5]`.

4. **Retrieving Original Indices**:
   - A list comprehension retrieves the original indices corresponding to these sorted unique energies:
     - For `energy = 7.1`, the index is `3`.
     - For `energy = 8.2`, the index is `1`.
     - For `energy = 10.5`, the index is `0`.
   - Thus, `unique_sorted_indices` becomes `[3, 1, 0]`.

5. **Return Statement**:
   - The function returns the list `[3, 1, 0]`.

assert unique_sorted_indices([10.5, 8.2, 10.5, 7.1, 8.2]) == [3, 1, 0]

---

**Code and Output:**

```python
from typing import List
def unique_sorted_indices(energies: List[float]) -> List[int]:
    energy_dict = {}
    for idx, energy in enumerate(energies):
        energy_dict.setdefault(energy, idx)
    sorted_unique_energies = sorted(set(energies))
    unique_sorted_indices = [energy_dict[energy] for energy in sorted_unique_energies]
    return unique_sorted_indices
assert unique_sorted_indices(??) == [3, 1, 0]
```

---

**Backward Monologue:**

```
### Understanding the Function
The function `unique_sorted_indices` processes a list of floating-point numbers (`
energies`) to produce the indices of the unique values sorted in ascending order. Here's
how it works:
1. **Initialization**: An empty dictionary `energy_dict` is created.
2. **Dictionary Population**: The function iterates through `energies` using `enumerate`,
 which provides both the index (`idx`) and the value (`energy`):
    - It uses `setdefault` to record the index for each unique energy value found in the
list. If the energy value already exists in the dictionary, it does not update the index
(thus only the first index is stored).
3. **Sorting Unique Energies**: A set of unique energy values is created from the list,
which is then sorted to create `sorted_unique_energies`.
4. **Index Retrieval**: A new list of indices is constructed by retrieving the original
indices from `energy_dict` corresponding to the sorted unique energies.
5. **Return Value**: The function returns a list of indices corresponding to unique
energy values in sorted order.

### Key Properties and Constraints
- The input list may contain duplicate energy values, but only the first occurrence's
index will be recorded in `energy_dict`.
- The output will be ordered according to the sorted unique energy values.
- The length of the output list will equal the number of unique energy values in the
input list.

### Reverse Deduction
Given the output `[3, 1, 0]`, we need to determine what `energies` might have been to
produce this result:
- **Behind the Result**:
   - `energy_dict` must map some unique energies to these indices.
   - The first unique energy corresponds to index `3`, the second unique to `1`, and the
third unique to `0`.
- **Energy Values**:
   - Suppose the values in the `energies` list that correspond to these indices are:
     - `energies[3]` -> first unique energy (let's call it `e1`)
     - `energies[1]` -> second unique energy (let's call it `e2`)
     - `energies[0]` -> third unique energy (let's call it `e3`)
- **Expected Ordering**:
   - In the sorted list of unique energies, `e1` must be the smallest, followed by `e2`,
and then `e3`.
- **Choice of Values**:
   - A simple set of values array can be created. We must ensure that:
     - The energy at index `3` is less than the energy at index `1` and both should be
less than the energy at index `0`.

Let's select:
- `energies[3] = 1.0` (first unique, smallest value)
- `energies[1] = 3.0` (second unique, middle value)
- `energies[0] = 5.0` (third unique, largest value)

To ensure proper indexing and repetition, we can fill the rest of the list with
duplicates of such values, for instance:
```python
energies = [5.0, 3.0, 5.0, 1.0]
```
With this arrangement, we maintain the index mapping:
- `5.0` at index `0` (first occurrence),
- `3.0` at index `1` (first occurrence),
- `1.0` at index `3` (first occurrence).

[/MONOLOGUE]
[ANSWER]
assert unique_sorted_indices([5.0, 3.0, 5.0, 1.0]) == [3, 1, 0]
[/ANSWER]
```

## H  Task-specific Prefix

We append task-specific prefixes to the training samples to guide the model to perform different types
of semantic reasoning.

---

**Task:** Natural Language to Code

---

**Prefix:**

```
    """
    You are an exceptionally intelligent coding assistant that consistently delivers
    accurate and reliable  according to <NL_Description>

    <NL_Description>
    {prompt}

    
    {completion}
    """
```

---

**Task:** Forward Monologue

---

**Prefix:**
```
    """
    Simulate the Execution: You are given a Python function and an assertion containing a
    function input. Complete the assertion containing the execution output corresponding to
    the given input in [ANSWER] and [/ANSWER] tags.
    {prompt}
    {completion}
    """
```

---

**Task:** Forward Monologue

---

**Prefix:**
```
    """
    Deduce the Semantic Constraints: You are given a Python program and its expected output.
    Find one input such that executing the program with the input leads to the given output.
    Complete the assertion with one such input in between [ANSWER] and [/ANSWER].
    {prompt}
    {completion}
    """
```

---

**Task:** Debug and Self-refine

---

**Prefix:**
```
    """Debug and Refine the Code: You are given a a problem to be implemented in Python, and
    a buggy code that tries to solve the problem but fails the test case.
    You should firstly simulate the execution with the buggy code and the failed test to
    identify the root cause of the failure.
    Then, you should fix the bug and wrap the refined code in between [Refined] and [/Refined
    ].
    {instruction}
    {response}
    """
```

---

# I  Baseline Trace Formats

We present the baseline traces formats as we discussed and compared in Section 6.2.

---

**Source Code and Test Case**

```
from typing import List # [L2]

def unique_sorted_indices(energies: List[float]) -> List[int]: # [L5]
    energy_dict = {}    # [L6]
    for idx, energy in enumerate(energies): # [L7]
        energy_dict.setdefault(energy, idx) # [L8]
    sorted_unique_energies = sorted(set(energies))  # [L9]
    unique_sorted_indices = [energy_dict[energy] for energy in sorted_unique_energies] #
[L10]
    return unique_sorted_indices # [L11]

assert unique_sorted_indices([10.5, 8.2, 10.5, 7.1, 8.2]) == [3, 1, 0] # [L13]
"""
```

---

**Scratchpad**

---

```
from typing import List

def unique_sorted_indices(energies: List[float]) -> List[int]: # [INPUT] {"energies":
[10.5, 8.2, 10.5, 7.1, 8.2]} [/INPUT]
    energy_dict = {}    # [STATE] {"energy_dict": {}} [/STATE]
    for idx, energy in enumerate(energies): # [STATE] {"idx": 0, "energy": 10.5} [/STATE
][STATE] {"idx": 1, "energy": 8.2} [/STATE][STATE] {"idx": 2, "energy": 10.5} [/STATE][
STATE] {"idx": 3, "energy": 7.1} [/STATE][STATE] {"idx": 4, "energy": 8.2} [/STATE]
        energy_dict.setdefault(energy, idx) # [STATE] {"energy_dict": "{10.5: 0}"} [/
STATE][STATE] {"energy_dict": "{10.5: 0, 8.2: 1}"} [/STATE][STATE] {"energy_dict":
"{10.5: 0, 8.2: 1, 7.1: 3}"} [/STATE]
    sorted_unique_energies = sorted(set(energies))  # [STATE] {"sorted_unique_energies":
[7.1, 8.2, 10.5]} [/STATE]
    unique_sorted_indices = [energy_dict[energy] for energy in sorted_unique_energies] #
[STATE] {"unique_sorted_indices": [3, 1, 0]} [/STATE]
    return unique_sorted_indices # [OUTPUT] [3, 1, 0] [/OUTPUT]
```

**NeXT Scratchpad**

```
from typing import List

def unique_sorted_indices(energies: List[float]) -> List[int]: # [INPUT] {"energies":
[10.5, 8.2, 10.5, 7.1, 8.2]} [/INPUT]
    energy_dict = {}    # [STATE-0] {"energy_dict": {}} [/STATE-0]
    for idx, energy in enumerate(energies): # [STATE-1] {"idx": 0, "energy": 10.5} [/
STATE-1][STATE-3] {"idx": 1, "energy": 8.2} [/STATE-3] ... [STATE-8] {"idx": 4, "energy":
 8.2} [/STATE-8]
        energy_dict.setdefault(energy, idx) # [STATE-2] {"energy_dict": "{10.5: 0}"} [/
STATE-2][STATE-4] {"energy_dict": "{10.5: 0, 8.2: 1}"} [/STATE-4][STATE-7] {"energy_dict
": "{10.5: 0, 8.2: 1, 7.1: 3}"} [/STATE-7]
    sorted_unique_energies = sorted(set(energies))  # [STATE-9] {"sorted_unique_energies
": [7.1, 8.2, 10.5]} [/STATE-9]
    unique_sorted_indices = [energy_dict[energy] for energy in sorted_unique_energies] #
[STATE-10] {"unique_sorted_indices": [3, 1, 0]} [/STATE-10]
    return unique_sorted_indices # [OUTPUT] [3, 1, 0] [/OUTPUT]
```

**Concise Trace**

```
"""
[L5] [INPUT] {"energies": [10.5, 8.2, 10.5, 7.1, 8.2]} [/INPUT] [/L5]
[L6] {"energy_dict": {}} [/L6]
[L7] {"idx": 0, "energy": 10.5} [/L7]
[L8] {"energy_dict": "{10.5: 0}"} [/L8]
[L7] {"idx": 1, "energy": 8.2} [/L7]
[L8] {"energy_dict": "{10.5: 0, 8.2: 1}"} [/L8]
[L7] {"idx": 2, "energy": 10.5} [/L7]
[L8] [/L8]
[L7] {"idx": 3, "energy": 7.1} [/L7]
[L8] {"energy_dict": "{10.5: 0, 8.2: 1, 7.1: 3}"} [/L8]
[L7] {"idx": 4, "energy": 8.2} [/L7]
[L8] [/L8]
[L7] [/L7]
[L9] {"sorted_unique_energies": [7.1, 8.2, 10.5]} [/L9]
[L10] {"unique_sorted_indices": [3, 1, 0]} [/L10]
[L11] [OUTPUT] [3, 1, 0] [/OUTPUT] [/L11]
"""
```

