# OpenReview forum: "SemCoder: Training Code Language Models with Comprehensive Semantics Reasoning"
_NeurIPS.cc/2024/Conference — NeurIPS 2024 poster_

### Official Review · Reviewer_H1xj · 2024-06-17

**Soundness:** 2
**Presentation:** 3
**Contribution:** 3
**Rating:** 6
**Confidence:** 4

**Summary:**

This paper proposes SemCoder, a code LLM with code semantic-aware pre-training. The authors first use OSS-instruct to create a synthetic dataset PYX for model training, and a PYX-R dataset for training LLMs for debugging and repair. Then the authors train the SemCoder model with NL to code, forward monologue, and backward monologue tasks. Results on code generation datasets show that SemCoder outperforms a group of open-source LLMs and GPT-3.5 with fewer parameters. SemCoder also demonstrates better ability in execution reasoning.

**Strengths:**

+The authors successfully trained a code LLM, and proposed novel approaches for both training data collection and training task design.

+The forward and backward monologue is interesting and proven helpful.

**Weaknesses:**

-In Section 2, I cannot find the examples for operational and abstract semantics in Figure 1 (“the middle yellow box” and “the extreme right box”).

-The approach of collecting traces seemed limited to simple programs without external dependencies and types. I wonder if SemCoder can still benefit code generation/understanding tasks in real-world data.

**Questions:**

- When building the PYX dataset, how did the authors choose the seed snippets?
- Why do the authors choose DeepSeekCoder-6.7b as their backend model (instead of other code LLMs or its instruction-tuned version)?
- Can the authors discuss why SemCoder-S significantly outperforms SemCoder on HumanEval, but achieves similar results on the rest tasks?

**Limitations:**

The limitations are well-discussed. However, I'm still curious whether the proposed approach can be extended to real-world programs with more complicated dependencies and data types?

---

> ### Author Rebuttal · Authors · 2024-08-06
>
> Thanks for your thoughtful comments!
>
> ## Seed Snippets
>
> PyX only uses parsable code snippets as seeds (Section 3.1). This is a small improvement over OSS-Instruct which randomly samples consecutive lines from code files, based on the observation that parsable seeds are more likely to yield syntactically correct programs. To do this, we first parse existing programs into ASTs and then randomly select sub-trees and use the corresponding source code as seed. As a result, we see a small performance boost over OSS-Instruct in Appendix Table-6.
>
> ## SemCoder-S’s Performance
>
> We studied samples in Evol-Instruct, which we combined to train the SemCoder-S: it notably increases the diversity of instructions for code generation, incorporating a broader scope and customized requests. Intuitively, this enhances the model's instruction-following capabilities, benefiting the HumanEval performance. However, the dominance of math problems in MBPP makes it less beneficial by only following instructions.
>
>
> Also, Evol-Instruct includes very limited execution-related signals, such as execution results or traces. As a result, the resource to learn dynamic semantics of execution will be mostly from PyX, which is shared by both SemCoder and SemCoder-S to learn to reason about execution. Consequently, these two variants have comparable capabilities in the execution reasoning tasks.
>
> ## Backbone Model
>
> We choose non-instruction-tuned checkpoints as the base model to compare with instruction-tuning methods to illustrate the importance of learning code semantics. Among all ~7B base Code LLMs, where our computing resources could efficiently support the developing cost, DeepSeekCoder-6.7B-base was one of the most capable open-source checkpoints when we worked on this project. However, the approach introduced by SemCoder is model-agnostic, we re-train SemCoder with two more base models (CodeLlama-7b-Python & DeepseekCoder-V2-Lite-Base) to show that the programming and execution reasoning capabilities of different checkpoints can all be significantly improved by SemCoder’s proposed method.
>
> | Model | HEval(+) | MBPP(+) | CRUXEval-I | CRUXEval-O |
> |--|--|--|--|--|
> | CodeLlama-7b | 37.8 (35.4) | 59.5 (46.8) | 37.3/40.4 | 34.6/34.0 |
> | SemCoder-CL | **73.2 (67.1)** | **69.0 (57.1)** | **43.4/49.9** | **38.2/51.1** |
> | DS-V2-Lite | 49.4 (42.7) | 78.3 (65.1) |  39.4/51.4 | 44.1/46.4 |
> | SemCoder-DS-V2-Lite | **81.1 (74.4)** |**84.4 (71.4)** |  **49.6/51.7** | **47.8/60.1** |
>
> ### Operational and Abstract Semantics
>
> Due to space limitations, we could not show the complete forward monologue of the motivating example in Figure 1, but we copy it below for illustration.
>
> ```
> [L3]: `energy_dict` is initialized as an empty dictionary.
> [L4]: The `for` loop iterates over the `energies` list with their indices using `enumerate`.
> First Iteration (idx=0, energy=10.5):
>    - [L5]: `energy_dict.setdefault(10.5, 0)` stores the key-value pair `(10.5, 0)` in `energy_dict`.
> Second Iteration (idx=1, energy=8.2):
>    - [L5]: `energy_dict.setdefault(8.2, 1)` stores the key-value pair `(8.2, 1)` in `energy_dict`.
> Third Iteration (idx=2, energy=10.5):
>    - [L5]: Since `10.5` already exists in `energy_dict`, its index is not updated.
> Fourth Iteration (idx=3, energy=7.1):
>    - [L5]: `energy_dict.setdefault(7.1, 3)` stores the key-value pair `(7.1, 3)` in `energy_dict`.
> Fifth Iteration (idx=4, energy=8.2):
>    - [L5]: Since `8.2` already exists in `energy_dict`, its index is not updated.
> [L6]: `sorted(set(energies))` creates a sorted list of unique energies: `[7.1, 8.2, 10.5]`.
> [L7]: List comprehension `[energy_dict[energy] for energy in sorted_unique_energies]` retrieves the original indices corresponding to the sorted unique energies: `[3, 1, 0]`.
> [L8]: The function returns `[3, 1, 0]`.
> ```
>
> As we defined in Section 2, the operational semantics refer to the individual steps in the code execution. From the complete forward monologue, we could see the execution side effect for each step, including the multiple executions of the same line (e.g., [L5] in the loop), will be reasoned by SemCoder during training.
>
> The abstract semantics refer to the  high-level description of program behavior without concrete details. In Figure 1 “Bottom-up Monologue”, the model “deduces” the input step-by-step. Since it is in the reverse order of the natural execution, the deduction requires abstract reasoning, such as “The energy values are supposed to be stored in the dictionary, but nothing is stored”. This is in contrast to the operational semantics, where certain variables should be reasoned with concrete values (e.g., `energies is [7.1, 8.2, 10.5]`)
>
>
> ### SemCoder’s Extensibility
>
> The primary goal of SemCoder is to propose a recipe for learning comprehensive semantics of code with LLMs. To achieve this, we initially focused on collecting self-contained code snippets with built-in types, allowing for a controlled environment to effectively extract semantic signals. This design choice allows us to collect code as well as generate high-coverage unit tests,  achieving thorough execution and efficient tracing to expose comprehensive semantics.
>
> Conceptually, SemCoder is extensible to real-world complexity by scaling up the computing to consume retrieved contexts and dependencies. First, monologue reasoning is designed to be general, similar to the inner monologue process when humans think. Monologues can be constructed for many manual analyses over realistic software (e.g., repository-level rubber-duck debugging). Second, the monologue is more efficient and flexible at representing complicated types and structures, compared to Scratchpad or NeXT. For example, when a variable is a tensor or very complicated class, Scratchpad or NeXT has to represent the lengthy and redundant details while monologue could simply describe the attributes that directly contribute to the downstream task, such as the shape of the tensor rather than all numeric values in it.

---

### Official Review · Reviewer_MYsU · 2024-06-21

**Soundness:** 3
**Presentation:** 4
**Contribution:** 4
**Rating:** 7
**Confidence:** 4

**Summary:**

Recent advancements in Code Large Language Models have primarily focused on code completion, yet these models often lack the ability to comprehend deeper semantics, such as the execution effects and dynamic states of code. This paper presents a pioneering approach aimed at enhancing Code LLMs' understanding of complex semantic tasks such as debugging and program repair. The methodology integrates high-level functional descriptions, local execution effects of individual statements, and the overall input/output behavior into the training of Code LLMs. This integration bridges the gap between static code text and dynamic execution states. The authors introduce PYX, a curated code corpus of fully executable samples with functional descriptions and execution tracing. They train Code LLMs to simulate human-like debugging by writing code and articulating execution behaviors in natural language. This approach culminates in the creation of SEMCODER, a specialized Code LLM with 6.7 billion parameters. SEMCODER demonstrates competitive performance against GPT-3.5-turbo in code generation and execution reasoning tasks, achieving 81.1% on HumanEval (compared to GPT-3.5-turbo's 76.8%) and 54.5% on CRUXEval-I (surpassing GPT-3.5-turbo's 50.3%). Furthermore, the authors explore the effectiveness of SEMCODER’s monologue-style execution reasoning versus concrete scratchpad reasoning, revealing a smoother integration of semantics across multiple dimensions. The study also underscores the potential of leveraging learned semantics to advance the debugging and self-refining capabilities of Code LLMs.

**Strengths:**

+ Important area.
+ The authors integrate the code semantics into SEMCODER.
+ The performance is good.

**Weaknesses:**

- Limited to Code Generation.

**Questions:**

This paper introduces a novel strategy to train Code Large Language Models (Code LLMs) that significantly enhances their understanding of complex tasks such as debugging and program repair through the integration of high-level functional descriptions and execution effects. The authors have developed SEMCODER, a model that performs competitively with GPT-3.5-turbo in standard code generation.

- Limitation to Code Generation:

While SEMCODER excels at tasks related to code generation and execution reasoning, its application to other areas of software development, such as code comment generation, remains unexplored. I believe the code semantics may be helpful in these areas.

Overall, the idea of this paper is good. I advocate to accept this paper.

**Limitations:**

Please see Questions, thanks.

---

> ### Author Rebuttal · Authors · 2024-08-06
>
> Thanks for your supportive feedback! We are glad that you like our idea of training code LMs with comprehensive semantics. Here, we would like to clarify the scope of SemCoder discussed in the paper and its potential for more software engineering applications.
>
> ## SemCoder’s Scope
>
> SemCoder extends its capabilities beyond basic code generation. Specifically, it has demonstrated substantial proficiency in execution reasoning through static input and output prediction (Execution Reasoning Columns in Table-1), as well as debugging and iterative refinement (Table-2). SemCoder outperforms all baseline models with up to 2x more parameters and also approaches the performance of much larger models like GPT-3.5 in tasks other than code generation. This highlights the model's effectiveness in both generating syntactically correct code and in deeply understanding code semantics and execution effects
>
> ## SemCoder’s Potentials
>
> Though SemCoder is mainly evaluated by well-established benchmarks, there are several practical applications in software development where its capabilities can be highly beneficial.
>
> **Bug Localization with Forward Monologue**: SemCoder's forward monologue reasoning could be instrumental in static bug localization. By simulating the real execution of a program line by line and reasoning about the effects of each statement, SemCoder can identify where the actual program states diverge from the expected behavior described in the functional specifications. This approach mirrors the rubber-duck debugging practice, where verbalizing the problem helps in pinpointing the issue.
>
> **Input Prediction with Backward Monologue**: The input prediction capabilities are useful for generating test cases or seed inputs for mutation-based testing. Given a desired program state to reach, SemCoder can perform backward monologue reasoning to synthesize appropriate inputs that will lead to that state. This ability is critical for ensuring comprehensive test coverage and for creating effective test cases that target specific parts of the code.
>
> **Code Comment Generation and Documentation**: We appreciate the reviewer’s suggestion of this task! Although not explicitly evaluated as a task in our paper, the monologue reasoning itself is actually a line-level code comment with a focus on the execution effect: for example, in figure-1, the forward monologues regarding line-3&4 are accurate code comments. Since SemCoder is trained to reason about both high-level functional descriptions and detailed execution semantics, we believe it could be easily extended to generate insightful code comments and documentation when fine-tuned with more task-specific data.

---

> > ### Comment · Reviewer_MYsU · 2024-08-12
> >
> > Thanks. The paper is good. I keep my score at 7.

---

### Official Review · Reviewer_sURP · 2024-07-11

**Soundness:** 3
**Presentation:** 3
**Contribution:** 3
**Rating:** 6
**Confidence:** 4

**Summary:**

This paper addresses the limitations of Code Large Language Models in understanding deeper semantics such as execution effects and dynamic states. It introduces a novel training strategy to enhance Code LLMs with comprehensive semantics, including high-level functional descriptions, local execution effects, and overall input/output behavior. Using a clean code corpus called PYX, the approach trains Code LLMs to write code and reason about execution behaviors using natural language, mimicking human verbal debugging. The results demonstrate the effectiveness of integrating semantics from multiple dimensions and improving debugging and self-refining capabilities.

**Strengths:**

+ The idea of introducing different semantics to augment code LLMs is quite interesting.
+ Using various kinds of semantic information to enhance code LLMs is a commendable approach.
+ The experiments effectively demonstrate the model's capabilities across different aspects.

**Weaknesses:**

- Some descriptions lack clarity.
- There are no use cases demonstrating the model's usefulness.

1. In section 6.2, how did you fine-tune the base models? What are the data generated for NeXT and Scratchpad, including both forward and backward monologues? Line 331 only mentions the backward monologues.
2. In Table 2, what metrics are used for evaluating input prediction, especially how can you evaluate with your abstract input representations? Can you show some trace examples of backward prediction for different baselines in the table? It is not easy to understand why using abstract semantics works better. Some case studies would be helpful.
3. In Table 3, why was NeXT not selected as a baseline, given its focus on repair? What are the differences between CodeLlama with PYX-R and SEMCoder (-S)? Since the base models are different, why wasn’t SEMCODER fine-tuned on CodeLlama and Magicoder for fair comparisons?

**Questions:**

Please check the questions in the weaknesses part.

**Limitations:**

N.A.

---

> ### Author Rebuttal · Authors · 2024-08-06
>
> We appreciate the reviewer’s detailed comments and feedback!
>
> ## Baselines in Section 6.2
>
> We apologize for the confusion. We provide an example of baseline trace formats in Appendix H. Given that neither scratchpad nor NeXT provides replication packages, we re-implement the approach following their illustration in the paper.
>
> **Data Format**: For both scratchpad and NeXT, we trace the execution and construct a dictionary for the changed variables after each execution step. We wrap such a dictionary in between two special tokens `[STATE]` and `[/STATE]` to represent the change of the program state, and if a line is executed multiple times, we will concatenate these states by the execution order. Finally, we append execution states to corresponding code lines as comments, as suggested by NeXT. The differences between Scratchpad and NeXT are two: (1) NeXT improves Scratchpad by explicitly indicating the sequential order of program states with numbers (2) NeXT ignores the intermediate states within a loop and replace them with `...`
>
> **Fine-tuning for Input and Output Prediction**: For baselines, we finetune the same base LLM (deepseekcoder-6.7b-base) as SemCoder to predict input and output but with the intermediate execution traces using the corresponding format shown in Appendix H.
>
> The model input of all baselines is the same, as shown in the “Source Code and Test Case” section of Appendix H, containing the source code and one executable input. The target sequence for the output prediction will be the inlined program states as shown in “Scratchpad”, “NeXT Scratchpad”, and “Concise Trace” sections. For the input prediction, we will reverse the line order of the inlined program states as the target sequence: for example, the scratchpad for input prediction will be:
>
> ```
>     return unique_sorted_indices # [ OUTPUT ] [3, 1, 0] [/ OUTPUT ]
>     unique_sorted_indices = [ energy_dict [ energy ] for energy in sorted_unique_energies ] #
> [ STATE ] {" unique_sorted_indices ": [3, 1, 0]} [/ STATE ]
>     sorted_unique_energies = sorted(set(energies)) # [STATE] {"sorted_unique_energies": [7.1, 8.2, 10.5]} [/STATE]
>     …
> def unique_sorted_indices ( energies : List [ float ]) -> List [ int ]: # [ INPUT ] {" energies ":
> [10.5 , 8.2 , 10.5 , 7.1 , 8.2]} [/ INPUT ]
> ```
>
> ## Advantages of Abstract Semantics
>
> There are two main advantages of abstract semantics by design compared to scratchpad-like execution traces.
>
> First, it properly handles the uncertainty in the program execution. For example, in the above example of backward execution reasoning, the model needs to deduce `energies` from its sorted set `sorted_unique_energies`. The scratchpad-like trace provides only one example of `energies` while the candidates could be multiple. In contrast, the backward monologue describes the high-level property of `sorted_unique_energies`: “It should be a disordered list with at least one 10.5, one 8.2, and one 7.1”, which teaches the model about the side effect of `sorted(set(energies))` rather than simply predicting one candidate. Second, abstract semantics maintain the flexibility of reasoning about key properties of dynamic program states rather than redundantly predicting details. For example, when reasoning about a tensor operation, such as transpose, the abstract semantics will focus on the important attributes like the shape, while not replicating numeric values.
>
> ## CRUXEval-I Evaluation
>
> As we have introduced in Section 4.2.2 (line 226-230), though the backward monologue will reason about abstract semantics, we force the model to conclude one __concrete executable input__ that satisfies the constraints deduced from the backward monologue. Therefore, we could validate the correctness of the prediction by executing it and compare with the expected output, similar to the standard setting implemented by CRUXEval-I.
>
> ## NeXT as Baseline
>
> Unfortunately, NeXT does not provide public artifacts. NeXT is a complicated system, and reproducing it from scratch requires at least (1) their training data: 10,047 program repairs from MBPP-R (2) the pre-trained weights that initialize NExT: Palm-2-L (3) the implementation of their self-training loops. However, none of these are publicly released; without the data and the initial checkpoint, it is nearly impossible to reproduce the model’s capabilities for a direct comparison in our setting. In addition, their evaluation data: the 1,468 held-out repair tasks from MBPP-R and the full set of HeFix+ are not released, either, so we could not evaluate SemCoder using the same evaluation setting as theirs.
>
> The most doable comparison is to compare their “naturalized execution trace” with our proposed monologue reasoning, so we refer to NeXT figure-2 in the original paper to implement the inlined execution trace as a baseline in Table-2.
>
> ### SemCoder with CodeLlama
>
> Thanks for the suggestion! Due to the limited computing resource, we construct SemCoder only with DeepSeekCoder as the proof-of-concept. However, the approach is generic and model-agnostic. To illustrate such generalizability and perform a fair comparison with CodeLlama, we train SemCoder-CL using Codellama-python-7b as the base model, and below is the result:
>
> | Model | CodeGen-HEval(+) | CodeGen-MBPP(+) | CRUXEval-I | CRUXEval-O | SelfRef-HEval(+) | SelfRef-MBPP(+)|
> |----------|----------|----------|----------|----------|----------|----------|
> | CodeLlama | 37.8 (35.4) | 59.5 (46.8) | 37.3 / 40.4 | 34.6 / 34.0 | 66.9 (56.1) | 72.6 (68.3) |
> | SemCoder-CL | **73.2 (67.1)** | **69.0 (57.1)** | **43.4 / 49.9** | **38.2 / 51.1** | **82.3 (79.3)** | **78.3 (78.0)** |
>
> `CodeGen` is Code Generation performance in Table-1 and `SelfRef` is the Self-Refine performance in Table-3.
>
> We can see that the SemCoder method significantly improves CodeLlama’s capabilities in code generation, execution reasoning, and debugging&self-refine, illustrating the effectiveness of SemCoder.

---

### Official Review · Reviewer_GuSF · 2024-07-12

**Soundness:** 3
**Presentation:** 3
**Contribution:** 3
**Rating:** 6
**Confidence:** 3

**Summary:**

The paper proposes PyX, a training dataset focusing on preserving the semantics of sample programs, and SemCoder, a coding language model trained with PyX with better ability to capture program semantics. The construction of PyX dataset comprises three phases to ensure the code samples involved are correct and self-contained. The dataset also contains pairs of buggy code, faulty traces and patches to improve the ability of LLMs in terms of debugging and self-refining. Based on the PyX dataset, a coding LM SemCoder is trained with a technique called Monologue Reasoning to improve the ability of code reasoning and comprehension in forward and backward directions. Comprehensive experiments are conducted to evaluate the performance of SemCoder, showing the SOTA performance and self-refining ability of SemCoder, along with the effectiveness of the training techniques involved.

**Strengths:**

+ A novel and open-source dataset to tackle a widely concerned problem
+ A language model trained on the dataset with SOTA performance
+ Easy-to-understand and effective ideas and techniques for the training process
+ Clear presentation

**Weaknesses:**

+ Hard to explain how monologue reasoning improves the performance of common coding tasks
+ Limited intuition on the experimental results, e.g. how does SemCoder conduct self-refining

**Questions:**

+ During the construction of PyX, sample programs are filtered following certain criteria (e.g. Python built-in types only). However, user-defined types such as classes are crucial and frequently-used in real-world programs. Will this act impair the generalizability and applications of PyX and SemCoder? How can this gap be tackled if possible?
+ During the construction of PyX, a generator model is adopted to generate pairs of problems and solutions. Is there any potential bias that the model tends to generate outputs with certain limitations (e.g. low complexity, or high similarity to the training dataset of the model)? If so, to what extend does it eventually impair the generalizability of PyX?
+ It seems that the monologue reasoning technique (especially the forward one) is targeted at concrete inputs and outputs of programs. What about abstract inputs and outputs that conforms to certain specifications? Can the technique improve the summarization (of abstract inputs/outputs and specifications) ability of LLMs , and how? Such ability can be crucial under certain tasks such as documentation/specification generation.
+ How does monologue reasoning improve the code comprehension ability of LLMs? Does the effect of monologue reasoning rely on the explicit textual analysis in the outputs of LLM? More specifically, how does the improved code reasoning ability of SemCoder comes into play under common coding tasks such as code completion, especially when explicit textual analysis on input code is not allowed? I expect to see more analysis and intuitions on this regard.

+ Can the author provide more examples and intuitions on the debugging and self-refining process of SemCoder? What kind of information is given to the LLM as feedback? How does the LLM utilizes the information? To what extend can the LLM identify the root cause? How effective is the LLM in terms of fixing the bugs?
+ Also, can the debugging ability of SemCoder be adopted into tasks like program repair? Is it possible to conduct more thorough evaluations on this regard with benchmarks more targeted on APR?

**Limitations:**

See weakness

---

> ### Author Rebuttal · Authors · 2024-08-06
>
> We appreciate the detailed reviews and questions!
>
> ## Self-contained Programs with Built-in Types
>
> The primary goal of SemCoder is to propose a recipe for learning comprehensive semantics of code with LLMs. To achieve this, we initially focused on collecting self-contained code snippets with built-in types, allowing for a controlled environment to effectively extract semantic signals. This design choice allows us to collect code as well as generate high-coverage unit tests,  achieving thorough execution and efficient tracing to expose comprehensive semantics.
>
> However, SemCoder is a general approach and can be extended to code snippets with additional complexity. Existing context retrieval tools and program analysis techniques, such as program slicing, can be utilized to retrieve the necessary out-of-scope context, such as cross-file dependencies and user-defined types. When a complete context is provided to the model, the monologue reasoning is generally applicable, regardless of whether the code contains only built-in types or user-defined types. On the other hand, we acknowledge the difficulty of generating thorough test cases for complex programs, which remains an open challenge in software engineering. Therefore, we regard such augmentation of PyX dataset as a future work.
>
> ## Bias in Problem-Solution Generation
>
> The potential bias introduced by the generator model for creating problem-solution pairs is possible when freely prompting the LLM. We mitigate this bias by providing diverse seed code snippets as the prompt for LLM to generate diverse response autoregressively, inspiring the a varied sampling process. As a result, PyX samples cover distinct topics and scopes. We show the category distribution in Figure-5 of Appendix, where each category could contain hundreds of distinct topics. We will add a diversity discussion with more plots and details in the next revision.
>
> ## Extension to Abstract Input and Output reasoning
>
> Currently, SemCoder was trained with code samples accompanied with concrete inputs and outputs for two main reasons: (1) we target to expose the dynamic program states with real execution for code semantic learning, which requires concrete executable inputs (2) we could validate the correctness of monologue reasoning (though not perfectly) by comparing its concluded input/output with the real execution results.
>
> However, as we introduced in Section 4.2, the monologue is constructed with the flexibility of including both operational and abstract semantics, where the latter describes abstract specifications or properties. Though such abstractions are more dominant in the backward monologue than the forward monologue in PyX, it should be straightforward to strengthen such abstractions by constructing a training dataset with abstract specifications as input and output and fine-tune SemCoder to drive its focus towards abstract semantics.
>
> ## The Effectiveness of Learning Monologue Reasoning
>
> We would clarify that the LLM’s general code comprehension capability gets improved with monologue reasoning training, regardless of the presence of explicit textual analysis during the inference. There are two main pieces of empirical evidence.
>
> First, in Table-1, SemCoder achieves state-of-the-art performance in code generation among <15B code LMs, significantly outperforming our base model DeepSeekCoder-6.7B-base. The code generation evaluation does not explicitly perform monologue, so this improvement illustrates the monologue reasoning produces a more capable programming system as it not only learns to generate code but comprehensively understands its semantics. However, as we have discussed in the "Appendix A - Limitations and Future Work", we acknowledge that the explicit monologue might further improve the accuracy of code generation.
>
> Second, as illustrated in the caption of Table-1, we evaluate SemCoder on CRUXEval with two settings: direct prediction (without monologue reasoning) of input and output (left side of “/”) and prediction with monologue reasoning (right side of “/”). We notice that SemCoder achieves notably better performance even without explicit monologue, highlighting the overall improvement of execution reasoning capabilities.
>
> (Due to the character restriction, we answer the last two questions in the following comment)

---

> ### Author Response · Authors · 2024-08-06
>
> ## Debugging, Self-refining, and Program Repairing
>
> Here, we show an example of debugging:
>
> **Model Input**
>
> ```
> Debug and Refine the Code: You are given a <Prompt> that describes a problem to be implemented in Python, <Fault & Trace> that the buggy implementation could not resolve the problem and fails the <Failed Test>, and the corresponding failed execution is traced and attached to code lines as comments.
> You should debug to identify the root cause of its failure.
> Finally, fix the bug and wrap the refined code in between [Refined] and [/Refined].
> <Prompt>
> Implement a function that takes a list of potential energies, sorts them in ascending order, removes duplicates, and returns the indices of the unique sorted energies.
>
> Write a function `unique_sorted_indices(energies: List[float]) -> List[int]` that takes a list of potential energies as input and returns a list of indices corresponding to the unique sorted energies.
> <Fault & Trace>
>
> from typing import List
>
> def unique_sorted_indices(energies: List[float]) -> List[int]:
>     energies_with_indices = list(enumerate(energies))
>     sorted_energies_with_indices = sorted(energies_with_indices, key=lambda x: x[1])
>     unique_sorted_energies_with_indices = list(set(sorted_energies_with_indices))
>     indices = [index for index, _ in unique_sorted_energies_with_indices]
>     return indices
>
> # …Execution Trace…
> # …
>
> <Failed Test>
> assert unique_sorted_indices([10.5, 8.2, 10.5, 7.1, 8.2]) == [3, 1, 0]
> ```
> **Model output**
>
> ```
> The root cause of the failure in the provided code is that the implementation does not handle duplicate energies correctly. The code removes duplicates based on the entire tuple (index, energy) instead of just the energy value. This leads to incorrect indices being returned for unique energies.
>
> To refine the implementation:
> 1. After sorting the energies, iterate through the sorted list and keep track of unique energies in a separate list.
> 2. While iterating, also store the corresponding indices of unique energies in another list.
> 3. Return the list of indices corresponding to unique energies.
>
> def unique_sorted_indices(energies: List[float]) -> List[int]:
>     energies_with_indices = list(enumerate(energies))
>     sorted_energies_with_indices = sorted(energies_with_indices, key=lambda x: x[1])
>     unique_energies = []
>     unique_indices = []
>     for index, energy in sorted_energies_with_indices:
>         if energy not in unique_energies:
>             unique_energies.append(energy)
>             unique_indices.append(index)
>     return unique_indices
> ```
>
> Motivated by NeXT, we provide the problem description, faulty code and its execution trace, and the failed test to the model as input, and the model will first analyze the root cause of the bug, and then propose the fix plan, finally followed by the concrete fix.
>
> We recognize that evaluating the accuracy of debugging is tricky since it is described as a rationale in natural language explaining the localization, root cause analysis, and fixing plan altogether, which could not be automatically evaluated. Therefore, we choose to manually study a few cases. We observe that when SemCoder correctly fixes the program (e.g., the above case), the debugging rationale is mostly accurate in all components of its analysis. We will include more case studies and manual analysis in the revision of the paper to illustrate SemCoder’s effectiveness of debugging and program repair
>
> ## Relation to APR
>
> As shown in the above example the debugging and self-refine process includes bug localization, root cause analysis, refine planning, and the final fix, so it is quite similar to the setting of APR.. Considering the short context length of SemCoder, we found it challenging to directly evaluate SemCoder on repository-level APR benchmarks, such as Defects4J, which require lengthy contexts to analyze the bug.

---

### Decision · Program_Chairs · 2024-09-25

**Decision:**

Accept (poster)

**Comment:**

This work introduces a dataset of executable programs with execution traces and uses this dataset to train SemCoder, a 6.7B parameter code-specific LLM. Reviewers generally recognized the substantial contribution, strong performance, and various specific technical contributions such as the forward & backward monologue objective. The main weakness of note is the relatively narrow focus on code generation tasks (in contrast to other code-related task), which is in part due to the reliance on relatively simple, executable programs. This concern did not outweigh the positives noted earlier, leading to the decision to accept the paper.